



# Temperature and CCN sensitivity of orographic precipitation enhanced by a mixed-phase seeder-feeder mechanism

Julia Thomas[1], Andrew Barrett[1], and Corinna Hoose[1]

[1]Institute of Meteorology and Climate Research, Department Troposphere Research (IMK-TRO), Karlsruhe Institute of Technology, Karlsruhe, Germany

**Correspondence:** Julia Thomas (julia.thomas@partner.kit.edu) and Corinna Hoose (corinna.hoose@kit.edu)

**Abstract.** Orographic precipitation is a key driver of flooding in mountainous areas. This article investigates the microphysical response of orographic rainfall to perturbations of temperature and cloud condensation nuclei (CCN) concentration. The study is motivated by the increased water vapour capacity of the atmosphere in a warming climate and the increasing frequency of extreme rainfall events. A case study for the Cumbria flood in December 2015 is performed with sensitivities using a realization of the 'piggybacking' method implemented into a limited-area setup of the ICON model. A $6\,\%\,\mathrm{K^{-1}}$ enhancement of rainfall results for the highest altitudes, caused by a 'mixed-phase seeder-feeder mechanism', i.e. the interplay of melting and accretion. Total $24\,\mathrm{h}$ rainfall is found to increase by only $2\,\%\,\mathrm{K^{-1}}$, significantly less than the $7\,\%\,\mathrm{K^{-1}}$ increase in atmospheric water vapour. A rain budget analysis reveals that the negative temperature sensitivity of the condensation ratio and the increase of rain evaporation dampen the rainfall enhancement. Decreasing the CCN concentration speeds up the microphysical processing, which leads to an increase in total rainfall. At low CCN concentration the rainfall sensitivity to temperature is systematically smaller. It is shown that the CCN and temperature sensitivities are to a large extent independent (with a $\pm 3\,\%$ relative error) and additive.

## 1 Introduction

The increasing intensity and frequency of extreme precipitation events caused by climate change is a major concern (Pörtner et al., 2022). Orographically enhanced precipitation is linked to the risk of flooding in the vicinity of low mountain ranges (Houze, 2012). A detailed understanding of the cloud microphysical processes that cause orographic precipitation is crucial to understand the climate impacts on cloud physics and extreme rainfall.

An important relationship that dictates the atmospheric water vapour content is the Clausius-Clapeyron (CC) equation for saturation vapour pressure of water $e_{\mathrm{sat}}$. In the atmospheric temperature range of interest in this work, $e_{\mathrm{sat}}$ increases between $6.5\,\%\,\mathrm{K^{-1}}$ and $9\,\%\,\mathrm{K^{-1}}$. As shown in Fig. 1, the increase is stronger for lower temperatures, i.e. higher up in the atmosphere. The value of $\mathrm{d}\log(e_{\mathrm{sat}})/\mathrm{d}T$ for a given temperature is referred to as CC scaling throughout the paper. If there were no changes in cloud dynamics and microphysics, the total precipitation would increase by the same rate as total vapour inflow. Deviations from this assumption can either be due to changes in dynamics, in the rate of condensation (thermodynamic effect) or due to changes in the microphysical processing of cloud water. The thermodynamic effect originates in the temperature sensitivity





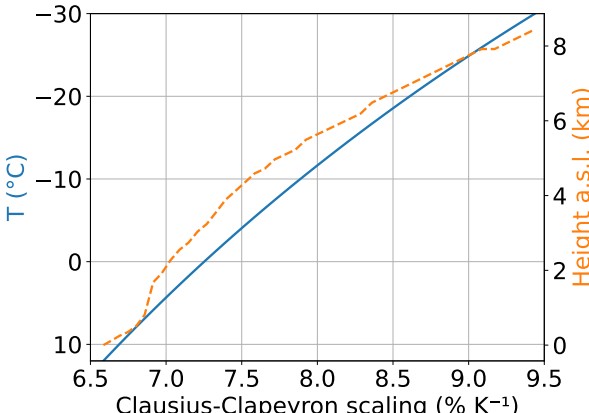

**Figure 1.** Clausius-Clapeyron scaling $d \log(e_{\text{sat}})/dT$ as function of temperature (solid blue) and as function of height a.s.l. (dashed orange) in the study region for the simulated case.

of the CC scaling: the relative increase in condensation is larger at high altitudes, according to the relative increase in water vapour. Siler and Roe (2014) showed that due to the temperature sensitivity of the moist adiabatic lapse rate, the increase in condensation is damped. It can lower the CC scaling by up to $4\,\%\,\text{K}^{-1}$ (Siler and Roe, 2014). Deviations from the CC scaling caused by cloud microphysics, which are characterized by the precipitation efficiency (PE) are the subject of this work. Relative humidity is assumed to by constant in a warmer climate, since enhanced evaporation balances the increased capacity of the

atmosphere to hold water vapour.

An important microphysical effect that enhances orographic precipitation is the seeder-feeder effect (Bergeron, 1965). This mechanism requires at least two layers of cloud of which the upper one is precipitating. The effect has been observed in many studies all over the globe (within nimbostratus clouds as well as related to orographic clouds) (e.g. Bergeron, 1965; Stow et al., 1991; Kunz and Kottmeier, 2006a) and is a main cause of orographically enhanced precipitation along the English

West coast (e.g. Browning et al., 1975; Smith et al., 2015; Hall, 2012), where low mountain ranges would otherwise not efficiently produce precipitation. The orographic cloud serves as 'feeder' cloud which is washed out by falling hydrometeors released from a 'seeder' cloud aloft. The upper cloud can as well be formed orographically. In most cases, however, it belongs to a layer of nimbostratus that usually forms along the warm front of a mid-latitude cyclone. One important feature of the mechanism is that the liquid water content (LWC) in the lower cloud can be continuously replaced by the low level moist

flow. The seeder-feeder mechanism does neither require the two cloud layers to be vertically separated nor do they have to be in the same thermodynamic state. Growth processes involving both frozen and liquid hydrometeors (i.e. aggregation, riming, collision-coalescence) can contribute to the precipitation enhancement process.

Riming and collision-coalescence become less efficient for smaller cloud droplets and a narrower size distribution, as is the case in environments with higher cloud condensation nuclei (CCN) concentrations (Alizadeh-Choobari, 2018). This leads to a

downwind shift of the surface rainfall distribution (Khain, 2009). If the raindrops are advected into the evaporation region at the





lee side of the mountain, the delayed onset of rainfall can lead to a decrease in total precipitation (Thompson and Eidhammer, 2014). In addition, smaller droplets can be lifted further up than larger droplets. This also favours advection into subsaturated regions. Meanwhile, droplets lifted above the freezing level can enhance graupel production by riming, which is a very efficient growth process. Which effect dominates highly depends on the synoptic conditions (e.g. whether convection is involved) and
on the mountain geometry (Kunz and Kottmeier, 2006a).

For the British Isles and mainland Europe, mid-latitude cyclones are the main drivers of wintertime precipitation (Douglas and Glasspoole, 1947), including extreme orographic precipitation events such as the Cumbria flood in December 2015. Similar orographic precipitation events, e.g. at the Norwegian West coast (Sandvik et al., 2018), over the Oregon Cascade Range (Garvert et al., 2007), the Southern Andes (Smith and Evans, 2007) as well as over the German Black Forest mountains (Kunz
and Kottmeier, 2006b) have been investigated in previous works.

At the British West coast, moist air flows moving westward over the Atlantic are lifted by low mountain ranges, producing orographic precipitation over and downstream of these coastal mountains. Preconditions for such events are often found within the warm sector of wintertime mid-latitude cyclones. These are

- prevailing fast low-level winds that lift the moist air mass over the mountain ridge (Browning et al., 1975);

- Froude numbers larger than 1 that prevent blocking and divergence of the flow (Kunz and Kottmeier, 2006a);

- roughly constant wind direction, such that the flow approaches the mountain range perpendicularly;

- a constant replenishment of moisture as source of orographic clouds and precipitation (Browning et al., 1975);

- comparably warm surface temperatures, such that low-level relative humidity (RH) is at least $80\,\%$ (Kunz and Kottmeier, 2006b). Only then low mountain ranges (up to $1\,\mathrm{km}$ height) are sufficient to lift the air mass above the lifting condensa-
tion level.

If ocean temperature exceeds the atmospheric temperature above the sea surface, evaporation is favoured and huge bands of moisture that transport water vapour from the Caribbean towards Europe can enhance the impact of the warm sector precipitation. These so-called atmospheric rivers are the main precursors of extreme precipitation events in Europe and are expected to become both more frequent, longer-lived and more enriched with water vapour in a warmer climate (Lavers and Villarini,
70   2015).

The Cumbria flood from 5th December 2015 caused severe flooding in the Lake District area in Northern England (Marsh et al., 2016). This event is chosen as a case study due to its unprecedented intensity. The rainfall totals exceeded the previous $24\,\mathrm{h}$ and $48\,\mathrm{h}$ UK records (Marsh et al., 2016). Surface temperatures exceeded $9\,^{\circ}\mathrm{C}$ even at night and strong winds with gust speeds up to $40\,\mathrm{m\,s^{-1}}$ were measured (Matthews et al., 2018; Marsh et al., 2016). Figure 2a shows the mean wind direction
during $6\,\mathrm{h}$ of the event. The moist flow approaches the Lake District mountains from the south-west with wind speeds up to $40$ $\mathrm{ms^{-1}}$ in the lower troposphere (Fig. 2b). Orographically induced updrafts of more than $1\,\mathrm{ms^{-1}}$ extend more than $4\,\mathrm{km}$ above sea level (a.s.l.). Correspondingly, downdrafts occur on the lee side of the hills.

The aims of this work are 1) to analyse the microphysical processes that enhance orographic rainfall in a mixed-phase cloud setting for the example of the Cumbria flood and 2) to analyse how these processes and their interaction change in a warming





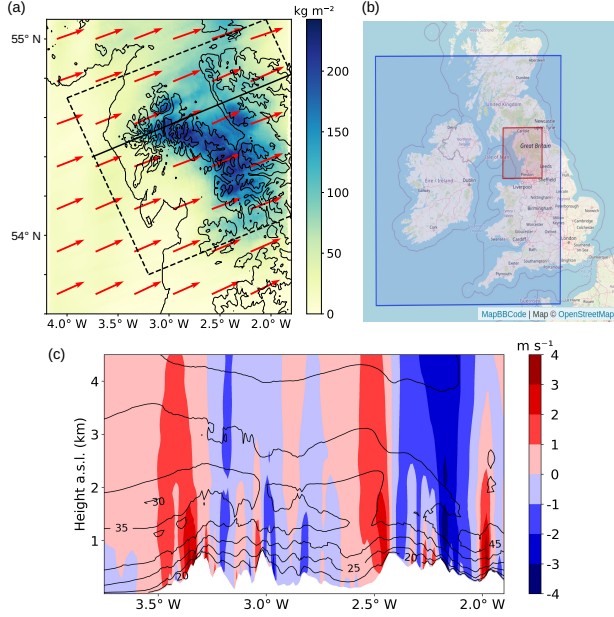

**Figure 2.** (a) 24 h accumulated surface rainfall (5 December 2015, 10:00 UTC to 6 December 2015, 10:00 UTC) on the inner nest. Black dashed polygon: evaluation domain (LAKEDISTR); solid black line: cross section used in (b) and in Sect. 3.2; red arrows: 6 h (see 3) average wind direction at 1.5 km a.s.l. (b) Map of outer and inner nest. (c) Up- and downdraft (red and blue shading) and horizontal wind speed (black contours) at 18:00 UTC along the cross section in (a).

atmosphere and with changes in the concentration of CCN. The analysis is based on convection-permitting simulations with the Icosahedral Non-hydrostatic model (ICON) developed by the German Weather Service (DWD) and the Max-Planck-Institute for Meteorology (Zängl et al., 2015) with an implementation of microphysical piggybacking (Grabowski, 2014). This method allows to evaluate microphysical sensitivities without the potentially confusing impacts of changes in the dynamics.

The article is structured as follows: Section 2 introduces the model setup, the piggybacking method as it is implemented in
ICON and the budget equation that is used to analyse the results quantitatively. The results of the simulation are presented and analysed in Sect. 3 and 4. Section 5 provides a discussion of the results and compares them with other studies. The results are summarized in Sect. 6.

## 2   Methods

### 2.1   Model configuration

The simulations are performed with ICON version 2.5.0 (Zängl et al., 2015) with an implementation of the piggybacking approach, i.e. the ability to perform additional calls to the cloud microphysics scheme using perturbed input parameters. ICON is run in limited area mode, with two domains (see Fig. 2(b)) nested into the standard ICON-EU grid (Reinert et al., 2020)





based on DWD analysis data. Boundary data is updated every 3 h for the simulations on the outer nest and every 15 min for the simulations on the inner nest. The outer nest has an average square equivalent edge length of 1.6 km and covers most of the British Isles. The dynamical timestep is 8 s. The simulation on the inner nest is initiated on 5 December 2015, 09:00 UTC (three hours after the start of the parent simulation) and is run for 27 h until 6 December 2015, 12:00 UTC. The inner nest extends from $-4.2\,°$ E to $-1.8\,°$ E latitude ($\approx 155$ km) and from $53.5\,°$ N to $55.1\,°$ N longitude (180 km) covering the Lake District area in northern England, as shown in Fig. 2(b). The triangular cells have an average square equivalent edge length of 445 m and the dynamical timestep is 2 s. The inner nest has 125 vertical levels extending up to 23 km a.s.l. At this grid spacing, a 3D turbulence scheme (Dipankar et al., 2015; Heinze et al., 2017) is used. Shallow convection is parameterized (Bechtold et al., 2008), but no deep convection scheme is used. The radiation time step (Mlawer et al., 1997; Prill et al., 2020) is 720 s. The two-moment cloud scheme is used (Seifert and Beheng, 2006) with CCN activation parameterized as a function of vertical velocity (Hande et al., 2016).

## 2.2 Piggybacking method

Piggybacking is a simple and computationally efficient method to separate microphysical and dynamical sensitivities (Grabowski, 2014). It is motivated by a challenge that all sensitivity studies with fully interactive models encounter: perturbations of microphysical parameters cause feedbacks on the thermodynamic state of the atmosphere (e.g. on temperature and buoyancy by latent heating) and consequently on the dynamics of the system, i.e. wind, pressure and static stability. Here we use piggybacking (a) to test the microphysical sensitivity of orographic rainfall to changes of thermodynamic conditions (here: changes in temperature) and (b) to investigate the response to changes in microphysical parameters, specifically the CCN number concentration.

The implementation of piggybacking applied in this study adds four sets of all microphysical prognostic variables to the model. Each set represents an individual simulation of cloud microphysics, driven by the same dynamic fields as the reference simulation. The cloud microphysics scheme is called five times per time step (each time updating one of the five microphysical variable sets). Therefore, each simulation results in five complete output variable sets with different microphysics but identical wind (see Fig. 2b) and pressure fields. In this work, the perturbed parameters are virtual potential temperature $\Theta_v$ and the surface CCN concentration $n_{\mathrm{CCN}}$. $\Theta_v$ instead of absolute temperature is used preserve static stability. $n_{\mathrm{CCN}}$ determines the vertical CCN profile (Hande et al., 2016). Except for $\Theta_v$ and specific humidity $q_v$ (in the temperature sensitivity experiments), the microphysical variable sets are initialised identically. The $\Theta_v$ and $q_v$ fields slowly diverge due to differences in calculated microphysical process rates resulting from the different (prognostic) temperature or (diagnostic) CCN concentration. Initially, $q_v$ is adapted to the perturbed value of $\Theta_v$ such that RH is identical to the reference simulation when the piggybacked simulation is initialized. This adjustment is motivated by the assumption that an increase in global temperature is accompanied by an increase in sea surface evaporation, such that RH will not change significantly in the future climate (Pörtner et al., 2022). For simplicity, perturbations of $\Theta_v$ are referred to as *temperature perturbations* throughout the paper.

Perturbations are chosen such that they resemble extreme but realistic deviations from the reference state. Warming and cooling scenarios are simulated by adding a constant offset of $\Delta\Theta_v = \pm 1$ K and $\Delta\Theta_v = \pm 3$ K to the virtual potential temper-





ature field in the microphysics scheme (and only there). To account for different degrees of atmospheric pollution, the initial value $n_{\text{CCN}} = 500$ cm$^{-3}$ has been rescaled by a factor of $0.1$ and $0.4$ to represent clean, maritime conditions and by a factor of $1.6$ and $3$ to represent polluted conditions. The resulting range of surface CCN concentration is thus $n_{\text{CCN}} = 50$ cm$^{-3}$, $200$
cm$^{-3}$, $500$ cm$^{-3}$, $800$ cm$^{-3}$ and $1500$ cm$^{-3}$. 25 simulations have been run in total, one for each combination of temperature and CCN perturbations. The simulations are denoted as PB-T-CCN, where T is the value of $\Delta\Theta_v$ in K and CCN is the value of $n_{\text{CCN}}$ in cm$^{-3}$. Thus, PB-0-500 denotes the reference simulation that provides the dynamics for all other simulations. An additional set of 4 simulations with temperature perturbations $\Delta\Theta_v = \pm 2$ K and $\Delta\Theta_v = \pm 4$ and $n_{\text{CCN}} = 500$ cm$^{-3}$ together with the five PB-T-500 simulations is referred to as PB-T ($-4$ K $\leq \Theta_v \leq +4$ K).

## 2.3   Analysis methods

### 2.3.1   Sensitivity decomposition

The function $\alpha_X(\Theta_v)$ with units $\%\,\text{K}^{-1}$ defined as

$$\alpha_X(\Theta_v) = \frac{1}{X}\frac{\partial X}{\partial \Theta_v} \cdot 100\ \% \tag{1}$$

is called the temperature sensitivity of quantity $X$. If there is no significant feedback, the dynamical, thermodynamic and
microphysical contributions to the total sensitivity can be linearly decomposed. For the sensitivity to total surface rainfall, $P$, this decomposition reads

$$\alpha_P(\Theta_v) = \frac{1}{P}\left.\frac{\partial P}{\partial \Theta_v}\right|_{\text{dyn}} + \frac{1}{P}\left.\frac{\partial P}{\partial \Theta_v}\right|_{\text{thermodyn}} + \frac{1}{P}\left.\frac{\partial P}{\partial \Theta_v}\right|_{\text{mphys}}. \tag{2}$$

Using the piggybacking method, the first term on the right hand side of Eq. (2), the dynamical contribution, vanishes since the dynamical fields are identical in each simulation. The term with index *thermodyn* corresponds to changes in rainfall caused by
the increase in water vapour inflow, $I$, and its condensation onto cloud droplets and deposition onto ice or snow ($C$), referred to as thermodynamic contribution. The index *mphys* refers to changes in the processing of cloud condensate, $C$, and precipitation formation (microphysical contribution). Since this work focuses on rain formation, the important microphysical processes are those related to rain generation (autoconversion, accretion and melting) and rain removal (riming of raindrops onto snow or graupel, evaporation of raindrops).

### 2.3.2   Evaluation domain and time averaging

The black dashed contours in Fig. 2 outline the evaluation domain, which is aligned with the mean wind speed and is referred to as LAKEDISTR. Figure 3 shows the hourly rates of surface rainfall, averaged over LAKEDISTR. The 16 h period (continuous line) starting on 5 December 2015, 10:00 UTC includes the highest rainfall rates and ends before a rapid decrease in rainfall is observed (02:00 UTC). This time marks the passage of the cold front. The 16 h time span is used for all totals defined in
the following section. The 24 h period (dashed) starting from 10:00 UTC is used to calculate the accumulated surface rainfall in Sect. 3.1. For computational efficiency, cloud content and process rates in Sect. 3.2 to 3.4 are averaged over a shorter time period (13:00 to 19:00 UTC) from 5 min data output. The results are qualitatively the same as for the 24 h period (not shown).



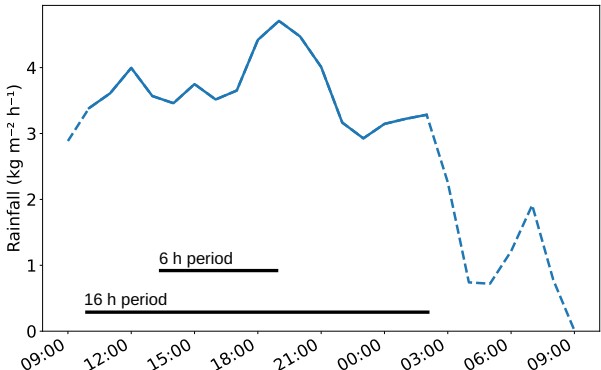

**Figure 3.** Hourly rainfall integrated over LAKEDISTR, starting on 5 December 2015, 10:00 UTC. The continuous line indicates the 16 h period used to calculate the totals in Sect. 4. Solid black lines: time periods used to average data.

### 2.3.3 Budget equation for total surface rainfall

The production of orographic rainfall can be decomposed into several phases: 1) the water vapour inflow, 2) the formation of an
orographic cloud and 3) the microphysical processing inside the cloud that eventually leads to formation and sedimentation of orographic rainfall. The amount of surface rainfall $P$ is related to the total water vapour inflow $I$ by the dimensionless drying ratio

$$DR = \frac{P}{I}. \tag{3}$$

$I$ is the integrated water vapour flux through the south-western boundary of LAKEDISTR, averaged over the 16 h period
shown in Fig.3. All totals introduced in this section are given in kg, such that the respective efficiencies are dimensionless. The following considerations are adapted from Kirshbaum and Smith (2008).

After the moist air entered LAKEDISTR (phase 1), it is forced to ascent by the mountain barrier. After saturation is reached, a part of it forms liquid (or ice) condensate (phase 2). The total amount of cloud condensate $C$ created this way is related to $I$ by the condensation ratio

$$CR = \frac{C}{I}. \tag{4}$$

The fraction of cloud condensate that is converted into rain and reaches the surface (phase 3) is called precipitation efficiency

$$PE = \frac{P}{C}. \tag{5}$$

With the three efficiencies $DR$, $CR$ and $PE$ defined, $P$ can be expressed as

$$P = I \cdot DR = I \cdot CR \cdot PE. \tag{6}$$





The temperature sensitivity of total surface rainfall (cf. Eq. 2) can now be written as follows:

$$\alpha_P = \frac{1}{P}\left.\frac{\partial P}{\partial \Theta_v}\right|_{\text{dyn}} + \frac{1}{I}\frac{\partial I}{\partial \Theta_v} + \frac{1}{CR}\frac{\partial CR}{\partial \Theta_v} + \frac{1}{PE}\frac{\partial PE}{\partial \Theta_v}$$

$$= \alpha_{P,dyn} + \alpha_I + \alpha_{CR} + \alpha_{PE}$$

$$= \alpha_I + \alpha_{CR} + \alpha_{PE} \quad \text{(with piggybacking)} \tag{7}$$

If $CR$ and $PE$ were unchanged in the different temperature and CCN scenarios, the amount of surface rainfall would increase
proportionally to the amount of atmospheric water vapour in the inflow ($\alpha_I$). The leading question of this work is to what extent
the temperature sensitivity of $P$ deviates from the Clausius-Clapeyron scaling, i.e. how $CR$ and $PE$ change with temperature
and which processes are responsible for their change. Furthermore, the effect of changing the CCN concentration at different
temperature scenarios shall be investigated.

The microphysical processing within the cloud can further be decomposed by defining a rain generation total

$$G = \text{autoconversion} + \text{accretion} + \text{melting} \tag{8}$$

and a rain loss total

$$L = \text{rain riming} + \text{rain evaporation} \,. \tag{9}$$

If lateral in- and outflow and initial condensate are negligible, $P$ equals the difference $G - L$.

## 3  Results

### 3.1  Spatial distribution of surface rainfall

Figure 4 shows the 24 h accumulated rainfall on the inner nest for the reference and the $\pm 3$ K cooling and warming scenarios.
The area that experiences extreme rainfall ($> 200$ mm) increases as the atmosphere warms, while the distribution of light
rainfall ($< 100$ mm) appears mainly unchanged. Notably, the area around the mountain ridge experiences the strongest increase
in rainfall. To quantify the observed changes, Figure 5a shows the hourly rainfall rate $p$ inside LAKEDISTR as well as the rate
above 600 m ($p_{\text{ridge}}$) and below 150 m ($p_{\text{low}}$) for temperature deviations of $-4$ K up to $+4$ K from the reference state. $p$
increases gradually at a rate of $1.6\ \%\,\text{K}^{-1}$. In contrast, $p_{\text{ridge}}$ increases at $6.0\ \%\,\text{K}^{-1}$ (close to CC-scaling) whereas $p_{\text{low}}$
decreases at $-1.1\ \%\,\text{K}^{-1}$. Figure 5b shows that the values of $p_{\text{ridge}}$ are within the 90th and 95th percentile of total rainfall at all
grid cells inside LAKEDISTR and that $p_{\text{low}}$ changes with temperature similar to the 60th percentile. Both Fig. 5a and Fig. 5b
reveal that regions with already high rainfall rates experience the strongest increase in rainfall with warming, whereas regions
with low rainfall rates in the PB-0-500 simulation experience a gradual change with temperature or even decreasing rainfall
rates. It is noteworthy that the numbers shown in Fig. 5 depend on the choice of the integration domain and time period (not
shown), but the qualitative results do not. The processes causing the orographic rain enhancement are disentangled in the next
sections.





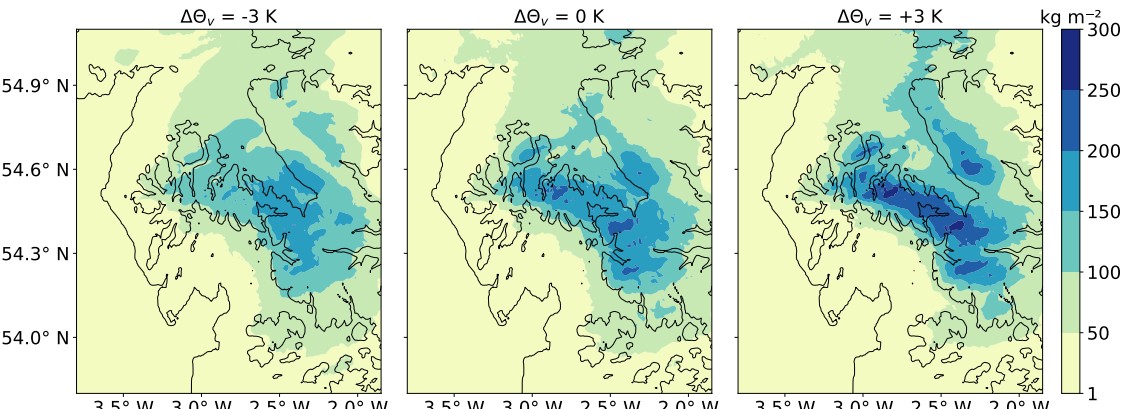

**Figure 4.** 24 h (5 December 2015, 10:00 UTC to 6 December 2015, 10:00 UTC) accumulated rainfall for three PB-T-500 simulations. Black contours indicate the orography at 1 m, 250 m and 500 m a.s.l.

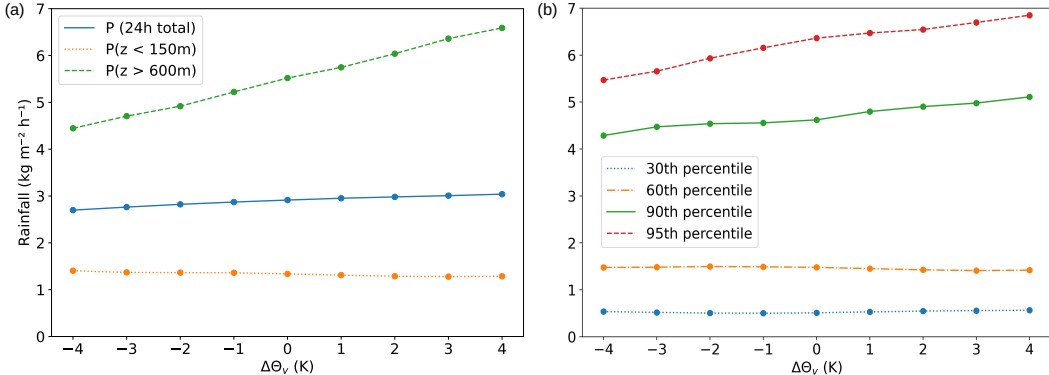

**Figure 5.** (a) Average 24 h rainfall rate evaluated inside LAKEDISTR ($p$ in solid blue, $p_{ridge}$ in dashed green, $p_{low}$ in dotted orange) for the PB-T simulations with $-4\,\mathrm{K} \leq \Delta\Theta_v \leq +4\,\mathrm{K}$. (b) 30th, 60th, 90th and 95th percentile for the PB-T simulations.

### 3.2  Cloud hydrometeor distribution along a vertical cross section

A comprehensive picture of the cloud distribution and the involved rain generating processes is given in Fig. 6 for three different temperature scenarios. Shown are filled contours of frozen (a) and liquid (c) cloud water content together with contours of melting and accretion rates (e). Below each contour plot, column integrated values of water content (b, d) or microphysical process rates (f) are displayed. Row (g) shows vertical profiles of rain generation and removal processes for each scenario. As in Fig. 4, results of the ±3 K cooling and warming scenarios are compared with the reference simulation. All values in Fig.

6 are averaged over the 6 h period starting at 13:00 UTC and evaluated along the vertical cross section indicated in Fig. 2, aligned with the mean wind. The cross section cuts through the Lake District as well as the subsequent mountain range (the Pennines).





**Figure 6.** (a) Filled contours of ice (purple), snow (red) and graupel (orange) content along the cross section shown in Fig. 2. (b) Column integrated values of the data presented in (a). (c) Filled contours of cloud water (green) and rain water (blue) along the same cross section. (d) Column integrated values of the data presented in (c). (e) Filled contours of graupel (orange), cloud (green) and rain water (blue) as in (a) and (c) together with contours of melting rate (red lines) and accretion rate (blue lines) in $[1,5,10,15]\,\mathrm{g\,m^{-3}\,h^{-1}}$. (f) Column integrated values of melting and accretion data presented in (e). (g) Vertical profiles of processes contributing to rain generation or rain removal. Values are evaluated inside LAKEDISTR. Columns correspond to the $\Delta\Theta_v = -3\,\mathrm{K}$ (left), $\Delta\Theta_v = 0\,\mathrm{K}$ (middle) and $\Delta\Theta_v = +3\,\mathrm{K}$ (right) simulations. All values are 6 h averages (from 5 December 2015, 13:00 UTC). Orography is outlined at the bottom of (a), (c), (e) and (g).





The uppermost row in Fig. 6 shows the distribution of ice, snow and graupel. Although the cold cloud extends up to $9\,\mathrm{km}$ a.s.l., it is initiated orographically by the first steep slope of the Lake District mountains at $30\,\mathrm{km}$ distance from the coast. The
cloud ice located between $6\,\mathrm{km}$ and $8\,\mathrm{km}$ a.s.l. is mostly unaffected by the temperature change, indicated in the comparison of the column integrated values in row (b). The amount of snow and graupel decreases as the temperature is increased from $-3\,\mathrm{K}$ to $+3\,\mathrm{K}$ w.r.t. the reference simulation, mostly due to the rise of the melting level from just below $1.5\,\mathrm{km}$ in the cooling case to $3\,\mathrm{km}$ in the warming case. The additional liquid condensate in the warming scenario then contributes to the liquid cloud content, as can be deduced from Fig. 6d. The replacement of frozen water content by liquid has two opposite effects on
rain production: enhancement of warm rain production via collision-coalescence and reduction of cold rain formation via the Wegener-Bergeron-Findeisen process and riming.

The liquid water path (LWP) has maxima above each ridge (Fig. 6d). In the cooling scenario, the liquid cloud water shown in Fig. 6c extends far downstream into the valley, while it is partially evaporated and partially converted into rain water above the valley in the warming scenario. The LWP at the eastward ridge is largely unchanged in the three temperature scenarios.
This suggests that the additional moisture contained in a warmer atmosphere is washed out before it is able to be advected downstream. A comparison of the competing effects - washout of cloud water and evaporation of rain - is made in Sect. 4.

### 3.3 Distribution of rain production processes

To determine which rain generation effect dominates and how atmospheric warming affects the interplay of cold and warm rain production, a detailed look at the associated microphysical processes is necessary. Figures 6e and 6f display average rates
of melting and accretion (i.e. collision-coalescence between cloud droplets and raindrops). These two processes quantify the main contributions to cold and warm rain production, respectively. Condensational growth of droplets as well as cloud droplet - cloud droplet collection (autoconversion) are negligible warm rain processes compared to the accretion of cloud droplets by raindrops. In the warming and cooling scenarios, the melting region is lifted or lowered according to the melting level. The two maxima of vertically integrated melting rates (Fig. 6f) decrease with temperature and are additionally shifted upwind in
the warming case. In contrast, the amount of accretion increases significantly with temperature, with two distinct accretion maxima over each first peak of the mountain range. As the liquid cloud top extends further upwards in the warmer scenarios, the accretion region does too, thereby increasing the value of the accretion maxima, without changing their location. In line with the liquid cloud water distribution, there is almost no accretion occurring over the valley.

The observed occurrence of melting and accretion can be explained considering typical time scales of cold and warm rain
processes. Melting of graupel, which is the dominating cold rain contribution, requires riming of liquid water droplets onto graupel or freezing of rain. The relatively long chain of processes, leaving time for horizontal advection caused by wind speeds of $30\text{-}40\,\mathrm{m\,s^{-1}}$ (Fig. 2b), leads to the observed distribution of melting with a maximum located far downwind of the mountain peaks. Accretion, on the other hand, is most effective in the region of maximum LWC. The altitude of maximum LWC is close to the mountain top (between $1\,\mathrm{km}$ and $2\,\mathrm{km}$ a.s.l. in the reference and warming scenario), such that most of the rain produced
in that region is deposited around the peaks. The plateau-like pattern downstream from the first accretion maximum is caused by the accretion of cloud droplets by melted graupel particles from the mixed-phase cloud.



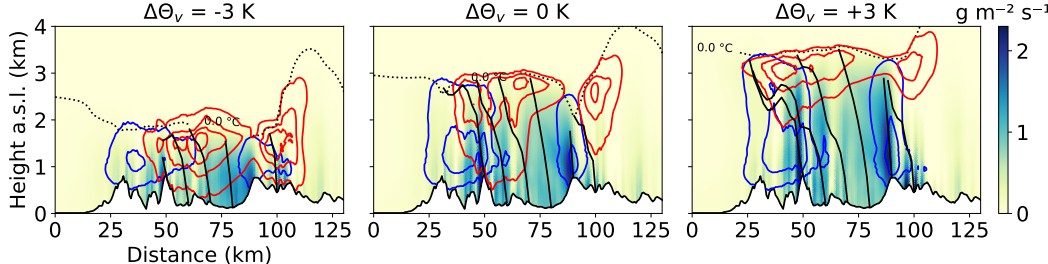

**Figure 7.** Rain water flux (shading) and raindrop trajectories (black) at 18:30 UTC (shading) together with contours of 6 h averaged melting (red) and accretion rates (blue) in [1,5,10,15] $\mathrm{g\,m^{-3}\,h^{-1}}$. The panels show the $\Delta\Theta_v = -3$ K (left), $\Delta\Theta_v = 0$ K (middle) and $\Delta\Theta_v = +3$ K (right) simulations. Orography is displayed at the bottom.

The vertical profiles in Fig. 6g show that melting of graupel contributes most to the rain generation budget. Autoconversion is the least important process and is highest in the warming scenario, where more cloud water is available. Both riming and melting decrease with warming and their maximum is lifted according to the melting level. Rain evaporation increases as

the total rain generation, i.e. the sum of autoconversion, accretion and melting, increases. Accretion increases strongly in a warming atmosphere, as the melting level rises and the liquid cloud layer extends vertically. A second maximum in the vertical profile of the accretion rate forms slightly below the melting level, visible in the reference and warming scenarios. The maxima correspond to levels of high LWC (horizontally averaged).

### 3.4 Raindrop trajectories

Figure 7 shows estimated raindrop trajectories starting at the melting level together with the mean rain water flux between 13:00 UTC and 19:00 UTC. The trajectory endpoints are chosen every 10 km starting at 50 km distance from the coast. The rain water flux is calculated from rain water mixing ratio $qr$ weighted with mass mean fall velocity in each grid cell, which is calculated as the sum of the updraft $w$ and the terminal velocity used in the microphysics scheme of the model (Seifert and Beheng, 2006). The terminal velocity is parameterized as a power law for the mean raindrop mass ($qr/qnr$). The fall velocity

than reads

$$v_{\mathrm{fall}} = w + 159.0 \cdot \left(\frac{qr}{qnr}\right)^{0.266} \cdot \left(\frac{\rho}{\rho_0}\right)^{0.5} \mathrm{m\,s^{-1}}, \tag{10}$$

with $\rho_0 = 1.255$ $\mathrm{kgm^{-3}}$ and air density $\rho$. Maxima of rain water flux are located above the highest peaks. The trajectories associated with these high rain water fluxes pass through regions where melting and accretion coincide or where accretion occurs below the melting layer. Such cases, where melting and accretion are spatially separated but occur along the trajectory

of falling hydrometeors yield the most efficient rain enhancement, e.g. for the 50 km trajectory in Fig. 6g (warming scenario).





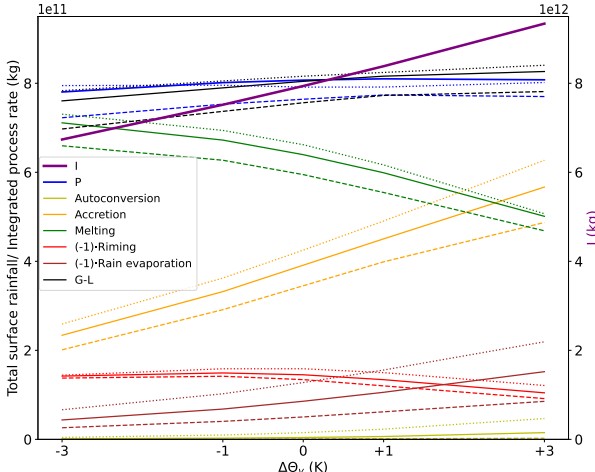

**Figure 8.** Processes contributing to rain generation (autoconversion, accretion and melting) and removal of rain (rain evaporation, riming onto graupel) for four temperature perturbations at three different CCN concentrations, $n_{\mathrm{CCN}} = 200\ \mathrm{cm}^{-3}$ (dotted), $500\ \mathrm{cm}^{-3}$ (solid) and $1500\ \mathrm{cm}^{-3}$ (dashed). The values are averaged over 16 h and integrated inside LAKEDISTR, as defined in Sect. 2.3. Black lines show the sum of the rain generation processes minus the sum of the removal processes ($G - L$). Blue lines show the 16-hour rainfall total $P$. The purple line shows the integrated water vapour inflow $I$ for all CCN concentrations. Values of $I$ are shown on the right axis.

## 4 Budget analysis

The first part of this section quantifies the relative importance of rain generation and removal processes. In the second part, the total rainfall is decomposed using the efficiencies introduced in Section 2.3. This allows to determine how efficiently water vapour is converted into surface rainfall and how the three phases of rain production - water vapour inflow, hydrometeor

formation and microphysical processing - change individually when the atmosphere warms. All totals are evaluated inside LAKEDISTR. Temperature sensitivities are given for the relative change between the $+3$ K warming scenario with intermediate CCN concentration (PB-plus3-500) and the reference simulation (PB-0-500), i.e. $\alpha_X = \alpha_X(3\,\mathrm{K})$. The temperature sensitivities obtained from all other PB-T-CCN simulations are listed in Appendix A in Table A1-A8.

### 4.1 Microphysical processes and surface rainfall budget

Figure 8 shows integrated values of all rain generation and loss processes as well as their sum together with total rainfall $P$ and total water vapour inflow $I$ obtained from the PB-T-CCN simulations. $I$ is one order of magnitude larger than $P$ and increases by $\alpha_I = 5.88\ \%\,\mathrm{K}^{-1}$ in the $+3$ K warming scenario (independent of CCN concentration). $P$ (16 h total) increases by only $\alpha_P = 0.03\ \%\,\mathrm{K}^{-1}$. The reason for this difference between $\alpha_I$ and $\alpha_P$ is examined in Sect. 4.2. The rain budget $G - L$ overestimates $P$ in the warming cases and slightly underestimates it in the cooling cases, with relative deviations of less than

$\pm 5\ \%$. This might be due to the exclusion of in- and outflow of rain water through the boundaries of LAKEDISTR.



**Figure 9.** Efficiencies in % as defined in Sect. 2.3 for the PB-T-CCN simulations. Values at three different CCN concentrations $n_{\mathrm{CCN}} = 200$ $\mathrm{cm}^{-3}$ (dotted), $500\,\mathrm{cm}^{-3}$ (solid) and $1500\,\mathrm{cm}^{-3}$ (dashed) are plotted over $\Delta\Theta_v$. Colours indicate condensation ratio (orange), precipitation efficiency (blue), and drying ratio (green).

Consistent with the vertical profiles shown in Fig. 6e, autoconversion increases with temperature ($\alpha_{\mathrm{auto}} = 87.29\,\%\,\mathrm{K}^{-1}$), but is a only a minor contribution to rain generation. It is more efficient when fewer CCN particles are available. Rain evaporation increases by $\alpha_{\mathrm{evap}} = 25.88\,\%\,\mathrm{K}^{-1}$, due to more rain water available, while riming of rain drops in the mixed-phase cloud region decreases, due to the lifted melting level and reduced graupel content. Melting and accretion dominate the rain gener-

ation budget. While accretion is significantly enhanced as the air gets warmer, increasing by $\alpha_{\mathrm{accr}} = 14.96\,\%\,\mathrm{K}^{-1}$, melting is reduced by $\alpha_{\mathrm{melt}} = -7.22\,\%\,\mathrm{K}^{-1}$ but remains almost as important as accretion even in the $+3$ K scenario. Reducing the CCN concentration from $500\,\mathrm{cm}^{-3}$ to $200\,\mathrm{cm}^{-3}$ (without perturbing $\Theta_v$) yields only a $0.93\,\%$ increase in total rainfall, although accretion is enhanced by $8.64\,\%$.

Altogether, the accretion enhancement (warm rain formation) cannot counteract the decrease in melting (a proxy for mixed-

phase precipitation formation) and the increased rain removal by evaporation ($25.88\,\%\,\mathrm{K}^{-1}$) in a warmer atmosphere. Therefore, the surface rainfall increases much less than the total water vapour inflow. In each temperature scenario, a cleaner atmosphere enhances all shown microphysical processes.

## 4.2   Precipitation efficiency, condensation and drying ratio

Following the definitions given in Sect. 2.3, the total rainfall integrated over LAKEDISTR can be written as the water vapour

inflow $I$ multiplied by the drying ratio $DR$, which is the product of condensation ratio $CR$ and precipitation efficiency $PE$. This decomposition helps to separate cloud microphysical processes (determining $PE$) from atmospheric thermodynamics (the main driver of $CR$) and to analyse their sensitivities individually. Figure 9 shows the efficiencies $DR$, $CR$ and $PE$ as functions of $\Delta\Theta_v$ for three values of $n_{\mathrm{CCN}}$. In the PB-0-500 case, $CR = 30\,\%$ of the inflowing water vapour is converted into cloud condensate. From the total condensate, $PE = 34\,\%$ reaches the ground as surface rainfall. The product $DR = CR \cdot PE$





shows that $10\,\%$ of the inflowing moisture sediments as rain to the surface (see also Fig. 8). The temperature sensitivities of these ratios are discussed in the following.

$I$ increases at a lower rate ($\alpha_I = 5.88\,\%\,\mathrm{K}^{-1}$) than the CC scaling in the temperature range of the Cumbria case ($6.5\,\%\,\mathrm{K}^{-1}$ to $7.5\,\%\,\mathrm{K}^{-1}$, Fig. 1), because the increase in absolute temperature is lower than the prescribed $3\,\mathrm{K}$ increase in $\Theta_v$. The partial derivative $\partial T/\partial\Theta_v$ ranges between $1$ (at the ground) and $0.75$ (at $8\,\mathrm{km}$ a.s.l.). Therefore, the actual temperature sensitivity of

$I$ is between $5.88\,\%\,\mathrm{K}^{-1}$ and $7.84\,\%\,\mathrm{K}^{-1}$, but is not calculated here explicitly.

The condensation ratio $CR = C/I$ decreases with increasing temperature ($\alpha_{CR} = -3.39\,\%\,\mathrm{K}^{-1}$). This is because the condensation total $C$, including condensation and deposition, increases with warming by only $1.89\,\%\,\mathrm{K}^{-1}$ (not shown). As mixed-phase processes play an important role in the Cumbria case, this value is the result of two counteracting effects with the same order of magnitude. The condensation rate obtained from saturation adjustment only, i.e. total increase in LWC from conden-

sational growth, yields a sensitivity of $5.44\,\%\,\mathrm{K}^{-1}$, similar to the change in $I$. In contrast, the total water vapour deposition to ice particles inside LAKEDISTR decreases by $-4.50\,\%\,\mathrm{K}^{-1}$. The negative sensitivity of depositional growth leads to the low value of the sensitivity of $C$ and explains the negative sensitivity of $CR$ (Fig. 9).

While rain generation is most efficient at high temperature perturbations, the precipitation efficiency $PE$ decreases by $\alpha_{PE} = -1.76\,\%\,\mathrm{K}^{-1}$ which is explained by the strong increase in rain loss ($3.74\,\%\,\mathrm{K}^{-1}$, Fig. 8), driven mainly by enhanced

evaporation. That means, the total amount of condensate (with an increased liquid fraction) is converted to rain less efficiently in a warmer atmosphere. As a result, the drying ratio decreases by $\alpha_{DR} = -4.97\,\%\,\mathrm{K}^{-1}$.

The condensation ratio $CR$ is larger for high CCN concentration, but has the same temperature sensitivity (see Table A6). This is due to enhanced condensation for high CCN concentration over the second mountain range. Since less water is washed out of the cloud in the polluted scenario, more moisture is advected downstream and can be condensed again, increasing $C$.

The precipitation efficiency $PE$ on the other hand is lower in the polluted scenario (Table A7), due to smaller droplets and reduction of the accretion efficiency. Furthermore, smaller droplets are more easily evaporated, once they are advected to the lee of the mountain. The two ($CR$ and $PE$) effects counteract each other, such that the CCN sensitivity of the drying ratio $DR$ is lower than the individual sensitivities of $CR$ and $PE$. However, the $PE$ sensitivity dominates, such that $DR$ decreases with increasing CCN concentration consistently in all temperature perturbation scenarios (Table A8).

### 4.3 Separating sensitivities in simulations with combined temperature and CCN perturbations

The total rainfall produced in the simulations with combined CCN and temperature piggybacking (PB-T-CCN) is analysed in this section. The 25 simulations are all driven by the dynamic fields of the reference simulation PB-0-500, with which the PB-T-CCN simulations are compared. The aim of this analysis is to (a) estimate the relative importance of changing CCN concentration compared to an increase in temperature and (b) identify whether the changes in $P$ in the PB-T-500 and PB-0-

CCN simulations are linearly independent.





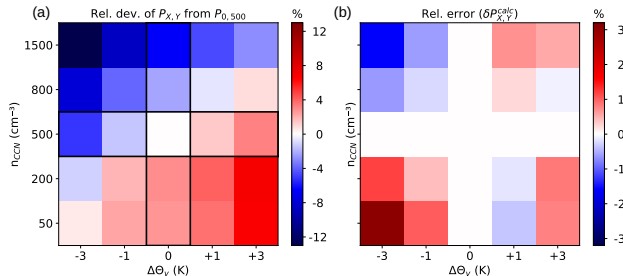

**Figure 10.** (a) Relative change in average 24 h rainfall in the PB-T-CCN simulation w.r.t. the reference simulation (PB-0-500). (b) Relative error of (by linear combination) calculated rainfall totals for the PB-T-CCN simulations.

**Table 1.** Temperature sensitivity $\alpha_P$ (in $\%\,\mathrm{K}^{-1}$) of 24 h rainfall $P$ for all $n_{\mathrm{CCN}}$.

| $n_{\mathrm{CCN}}$ (cm$^{-3}$) | $\Delta\Theta_v = -3\,\mathrm{K}$ | $\Delta\Theta_v = -1\,\mathrm{K}$ | $\Delta\Theta_v = +1\,\mathrm{K}$ | $\Delta\Theta_v = +3\,\mathrm{K}$ |
|---|---|---|---|---|
| 50 | 0.76 | 0.5 | 1.01 | 1.40 |
| 200 | 1.35 | 1.06 | 1.17 | 1.41 |
| 500 | 1.79 | 1.53 | 1.40 | 1.17 |
| 800 | 2.03 | 1.80 | 1.69 | 1.19 |
| 1500 | 2.40 | 2.24 | 2.22 | 1.48 |

Let $P_{X,Y}$ be the 24 h rainfall total in any PB-X-Y simulation, where $X$ denotes the temperature perturbation $\Delta\Theta_v$ in K and $Y$ the CCN concentration $n_{\mathrm{CCN}}$ in cm$^{-3}$ and $P_{0,500}$ is the 24 h rainfall total in the reference simulation (PB-0-500). Then

$$\Delta P_{X,Y} = P_{X,Y} - P_{0,500} \tag{11}$$

is the absolute difference in rainfall in any PB-X-Y simulation from the reference simulation. The individual contributions from the PB-T and PB-CCN simulations are then

$$\Delta P_X^{\Delta T} = P_{X,500} - P_{0,500} \tag{12}$$
$$\Delta P_Y^{\mathrm{CCN}} = P_{0,Y} - P_{0,500}. \tag{13}$$

Values for $\Delta P_{X,Y}$ (normalized by $P_{0,500}$) are shown in Fig. 10a. The relative deviations vary from $-13\,\%$ for the coolest and most polluted scenario to $+7\,\%$ in the warmest and cleanest scenario. Consistently, at all temperature perturbations, an increase of CCN concentration yields a decrease of $P$, and at all CCN perturbations, an increase in temperature results in an increase in $P$. Table 1 shows the temperature sensitivities $\alpha_P$ calculated for the four perturbations of $\Theta_v$ w.r.t. the reference simulation at fixed CCN concentration. Simulations run in more polluted scenarios ($n_{\mathrm{CCN}} = 800\,\mathrm{cm}^{-1}$ and $1500\,\mathrm{cm}^{-1}$) have systematically higher sensitivities than the simulations under cleaner conditions ($n_{\mathrm{CCN}} = 50\,\mathrm{cm}^{-1}$ and $200\,\mathrm{cm}^{-1}$). The sensitivities vary between $0.5\,\%\,\mathrm{K}^{-1}$ and $2.4\,\%\,\mathrm{K}^{-1}$ but are still much smaller than the CC scaling.



If feedback of temperature change on CCN concentration and vice versa are negligible, the PB-T and PB-CCN contributions can be used to calculate the rainfall totals as a linear combination

$$P_{X,Y}^{\text{lin}} = \Delta P_X^{\Delta T} + \Delta P_Y^{\text{CCN}} + P_{0,500} \,. \tag{14}$$

To analyse to what extent this sum deviates from the combined effect $P_{X,Y}$, the relative error

$$\delta P_{X,Y}^{\text{lin}} = \frac{P_{X,Y}^{\text{lin}} - P_{X,Y}}{P_{X,Y}} \tag{15}$$

is shown in Fig. 10b. The calculated rainfall deviates from the simulated rainfall by less than 3 %. In the warming scenarios, deviations are smaller and there is no systematic over- or underestimation. In the cooling scenarios, adding the individual contribution at reduced CCN concentration causes an underestimation of the produced rainfall, and doing so at increased CCN concentration yields an overestimation. However, the individual CCN and temperature contributions can be used to estimate the total rainfall for an arbitrary combined scenario at $\pm 3$ % accuracy. This finding supports piggybacking as a powerful method
to separate thermodynamic and microphysical sensitivities.

## 5   Discussion

In this section the results shown in Sect. 3 and 4 are discussed and compared with literature. In addition, limitations and possible extensions are discussed.

### 5.1   Interpretation and comparison

The interplay of cold and warm rain processes in the Cumbria flood case can be characterized as *mixed-phase* seeder-feeder mechanism. The rain enhancement is strongest when melted hydrometeors from the mixed-phase region serve as seeder particles that collect cloud droplets from the liquid (feeder) cloud region below. Cases in which the melting and accretion regions are vertically separated yield the most efficient rain enhancement, because accretion can efficiently occur throughout the full extent of the liquid cloud layer. The raindrops sediment at the downwind side of the hills that trigger the orographic cloud for-
mation. Drops created too far in the lee are evaporated. This effect is mainly responsible to dampen the rain enhancement, as also observed in Siler and Roe (2014). In a warmer atmosphere, melting happens earlier such that accretion below is enhanced not only due to the overall increase in LWC but also because more melted hydrometeors fall through the liquid cloud layer. Therefore, although total melting is reduced, the mixed-phase seeder-feeder effect leads to a strong enhancement of rainfall at locations close to the mountain ridge. However, the surface rain total increases less than the total water vapour inflow, mainly
due to the decrease of the condensation ratio and enhanced evaporation.

    Siler and Roe (2014) showed that the higher increase of condensation at high altitudes leads to a downwind shift of rainfall. This shift dominated in their idealized study and was not counteracted by the microphysical effect of increased liquid to frozen hydrometeor ratio in a warmer atmosphere that can lead to an upwind shift the distribution of precipitation. In this study, the faster processing of droplets indeed leads to an upwind shift of the rainfall pattern. A reason for the diverging results may be





that in Siler and Roe (2014) most of the orographic rainfall sedimented on the upwind facing side of the mountain, due to
lower wind speeds. In this study, the rainfall is deposited on the lee slope. In addition to faster processing, the rain water solely
produced by melting is evaporated over the valley in the warming scenario.

In terms of total precipitation, Siler and Roe (2014) found an increase in $P$ of $4.7\,\%\,\mathrm{K}^{-1}$ and Sandvik et al. (2018) found an
increase of $5\,\%\,\mathrm{K}^{-1}$, while in this study the 24 h rainfall increase varies between $0.5\,\%\,\mathrm{K}^{-1}$ and $2.4\,\%\,\mathrm{K}^{-1}$ depending on the
temperature perturbation and CCN concentration. This discrepancy in $\alpha_P$ may be due the choice of the integration domain as
discussed in Sect. 3.1 and due to the lower temperature sensitivity of $I$ in this study compared to others. The surface temperature
range in Siler and Roe (2014) and Sandvik et al. (2018) is comparable to this study (i.e. $T_{\mathrm{surface}} \approx 10\,^\circ\mathrm{C} - 15\,^\circ\mathrm{C}$). Comparable
with the $6.0\,\%\,\mathrm{K}^{-1}$ increase of $P$ above 600 m a.s.l found in this study, Sandvik et al. (2018) found a stronger increase in
precipitation at high altitudes. Above $650\,\mathrm{m}$ a.s.l. $P$ increases by $6.4\,\%\,\mathrm{K}^{-1}$ and below 150 m a.s.l. $P$ increases by $2.3\,\%\,\mathrm{K}^{-1}$
(Sandvik et al., 2018). However, in this study the amount of rainfall decreases at altitudes below 150 m. An intensification
of heavy orographic rainfall (sedimenting at high altitudes) at the expense of moderate and weak rainfall (sedimenting at low
altitudes) is also found in an ensemble study over Norway that used a regional climate model to simulate a future climate
scenario (Poujol et al., 2021).

The temperature sensitivities of the condensation and drying ratios agree qualitatively with previous studies, although the
magnitudes differ. In agreement with Sandvik et al. (2018) who found that $CR$ decreases by $3\,\%\,\mathrm{K}^{-1}$, the $\alpha_{CR} = -3.4\,\%\,\mathrm{K}^{-1}$
sensitivity found in this work dominates the sensitivity of $DR$. Beside the fact that moist air needs to be lifted higher up to reach
saturation in a warmer atmosphere the reduction of $CR$ is mainly caused by the $7.7\,\%\,\mathrm{K}^{-1}$ reduction of frozen hydrometeor
content. The total condensation $C$ increases by $5.4\,\%\,\mathrm{K}^{-1}$ in the PB-plus3-500 scenario, comparable to the results obtained in
the idealized study by Siler and Roe (2014), who found an increase in upstream condensation of $5.7\,\%\,\mathrm{K}^{-1}$. Together with the
slight reduction of $PE$, caused by the transition from cold to less efficient warm rain production and enhanced rain evaporation,
the negative sensitivity of $CR$ yield a total decrease in $DR$ of $\alpha_{DR} = 4.97\,\%\,\mathrm{K}^{-1}$. In fact, all studies discussed here find that
$DR$ decreases with temperature, mainly caused by the thermodynamic effect. Sandvik et al. (2018) found $DR$ to decrease by
$1.2\,\%\,\mathrm{K}^{-1}$ and Kirshbaum and Smith (2008) found $\alpha_{DR} = -3.1\,\%\,\mathrm{K}^{-1}$. Presumably, those values are less extreme than the
value obtained here of $-4.97\,\%\,\mathrm{K}^{-1}$ because the water vapour inflow in their simulations increased by $10\,\%\,\mathrm{K}^{-1}$ (Siler and
Roe, 2014) and $11\,\%\,\mathrm{K}^{-1}$ (Kirshbaum and Smith, 2008), whereas in this study it is only $\alpha_I = 5.88\,\%\,\mathrm{K}^{-1}$, which is partially
explained by the smaller change in actual temperature compared to the $\Delta\Theta_v$ offset.

Strong changes in CCN can modify the surface rainfall to a similar amount as the considered temperature changes. In our
setup with fixed dynamics, these changes are approximately linearly independent of the temperature changes of thermodynamic
and microphysical processes. At high CCN, the temperature sensitivity of surface precipitation is slightly higher than at low
CCN concentrations, but still small compared to CC scaling.

Despite the differences in numbers, the tendencies found in this study agree well with previous work on orographic pre-
cipitation in climate change. Total precipitation per event is expected to increase with temperature, but at a lower rate than
atmospheric water vapour. However, the mixed-phase seeder-feeder effect acts to focus the rainfall onto the highest elevations
and thus poses great risk for flooding in and around mountainous regions.



## 5.2 Limitations and potential solutions

The piggybacking approach used to conduct this sensitivity study proves powerful to test microphysical sensitivities in isolation. However, locking the atmospheric dynamics excludes a big part of physics that itself is affected by climate change. Changes of global circulation patterns can affect the size, pathway and intensity of atmospheric rivers and mid-latitude cyclones. These large scale changes in the location, frequency or dynamics of mid-latitude storms might be more important than local changes around certain mountain ranges (Siler and Roe, 2014; Shi and Durran, 2014). One approach to account for that problem could be to adjust $\Delta\Theta_v$ not to be a constant offset but by a 3-dimensional field based on the output of a regional climate model, similar to Poujol et al. (2021). This way, more realistic temperature perturbations could be applied, although the dynamics remain identical.

Previous studies did not use piggybacking and thus the comparison must be treated with care. However, those used for comparison here found low sensitivities of precipitation to changes in atmospheric dynamics (Sandvik et al., 2018; Siler and Roe, 2014; Kirshbaum and Smith, 2008).

Another limitation is the focus on a single case. This study thus lacks the more robust conclusions that could be derived from a larger statistical ensemble of various orographic rainfall events. As previously mentioned, the results depend strongly on the choice of the time period and evaluation domain. This is a general issue of single realistic studies, but it seems to be particularly challenging in the Cumbria case, due to its long duration and complex terrain. In order to generalize the findings, this study could be challenged by (a) performing further analyses of other extreme events in the same area, (b) splitting up the integration domain and the time interval and compare the sensitivities obtained from each sub-domain, (c) extending the analysis to other cases of extreme orographic precipitation. In general, coastal mountain ranges located at the West coast of continents, such as the Olympic Mountains in Washington (US), the Norwegian coastal mountains or the Aoraki National Park in New Zealand are suitable choices if the synoptic conditions - air temperature, wind speed and direction, static stability - and the mountain geometry are comparable. A comparison of extreme precipitation events in cases with and without mixed-phase clouds could reveal the importance of melting as well as the potential for rain enhancement in a liquid-only cloud setting. A sensitivity analysis focusing on the mountain geometry, e.g. re-scaling the orography, could help to generalize the findings.

## 6 Summary

This work analysed the temperature and CCN sensitivities of orographic rainfall embedded in a wintertime mid-latitude storm. To the authors' knowledge, it is the first study to apply piggybacking in sensitivity experiments of orographic rainfall.

A slight enhancement of $1.6\,\%\,\mathrm{K}^{-1}$ of 24 h rainfall was found in the simulations with perturbed $\Theta_v$. The strong deviation from the $7\,\%\,\mathrm{K}^{-1}$ CC scaling is due to the negative temperature sensitivity of the drying ratio. However, a $6.0\,\%\,\mathrm{K}^{-1}$ increase of rainfall at the mountain peaks was found, whereas rainfall at low altitudes decreased by $1.1\,\%\,\mathrm{K}^{-1}$. The intensification of rainfall around the mountain peaks was caused by strongly enhanced accretion in a warmer atmosphere together with the upwind shift of melting, such that both processes have an increased vertical overlap. This effect is termed mixed-phase seeder-feeder mechanism.





Analysis of non-dimensional efficiency measures showed that less efficient condensation and deposition of cloud condensate ($\alpha_{CR} = -3.4\,\%\,\mathrm{K}^{-1}$) in a warmer climate is mainly responsible for the fact that rainfall enhancement is much lower than the

increase in water vapour inflow ($\alpha_{DR} = \alpha_{CR} + \alpha_{PE} = -4.97\,\%\,\mathrm{K}^{-1}$). Enhanced lee side evaporation of rain water yields a slight decrease in precipitation efficiency ($\alpha_{PE} = -1.57\,\%\,\mathrm{K}^{-1}$).

Separating the temperature and CCN contributions to total increase in $P_{X,Y}$ showed that the individual contributions are independent of each other. Orographic rainfall is expected to increase in both warmer and cleaner environments. The precipitation increases are largest over the mountain peaks, where the precipitation totals are already the largest, by the mixed-phase

seeder-feeder mechanism. This implies that the risk for severe rainfall in mountainous regions via the seeder-feeder mechanism may increase in future.

*Code and data availability.* ICON model output and pre-processing scripts are available upon request and will be made available in a repository after final acceptance of this manuscript.

*Author contributions.* AB developed the piggybacking implementation for ICON. AB and JT performed the numerical simulations. JT

conducted the analyses and all co-authors contributed to the interpretation of the results. JT wrote the paper, with support from all co-authors.

*Competing interests.* The authors declare that none of them has any competing interests.

*Acknowledgements.* Funding from the German Research Foundation (Deutsche Forschungsgemeinschaft, DFG) through subproject "Evaluating and Improving Convection-Permitting Simulations of the Life Cycle of Convective Storms using Polarimetric Radar Data" under the programme SPP-2115 "PROM" and the Transregional Collaborative Research Center SFB / TRR 165 "Waves to Weather", and from the

European Research Council (ERC) under the European Union's Horizon 2020 research and innovation programme under grant agreement No 714062 (ERC Starting Grant "C2Phase") is gratefully acknowledged. This work was performed on the computational resource ForHLR II funded by the Ministry of Science, Research and the Arts Baden-Württemberg and DFG.

## Appendix A: Scaling parameters for different temperature and CCN scenarios

In the following, temperature sensitivities of the most relevant quantities analyzed in this study are shown analogous to Tab.

1. They are calculated as relative changes from the corresponding $\Delta\Theta_v = 0\,\mathrm{K}$ scenario for each value of $n_{\mathrm{CCN}}$, normalized by the respective temperature perturbation, as a proxy for the temperature sensitivity $\alpha_X$ in units of $\%\,\mathrm{K}^{-1}$. This way, each table contains $5 \cdot 4$ values.





**Table A1.** Temperature sensitivity (in $\% \, \text{K}^{-1}$) of average rain water content $RWC$ for all $n_{\text{CCN}}$.

| $n_{\text{CCN}}$ (cm$^{-3}$) | $\Delta\Theta_v = -3\,\text{K}$ | $\Delta\Theta_v = -1\,\text{K}$ | $\Delta\Theta_v = +1\,\text{K}$ | $\Delta\Theta_v = +3\,\text{K}$ |
|---|---|---|---|---|
| 50 | 11.68 | 13.46 | 14.13 | 13.91 |
| 200 | 12.27 | 13.96 | 14.44 | 14.42 |
| 500 | 12.91 | 14.63 | 14.54 | 13.58 |
| 800 | 13.19 | 14.93 | 14.81 | 13.20 |
| 1500 | 13.38 | 15.03 | 15.11 | 12.93 |

**Table A2.** Temperature sensitivity (in $\% \, \text{K}^{-1}$) of accretion for all $n_{\text{CCN}}$.

| $n_{\text{CCN}}$ (cm$^{-3}$) | $\Delta\Theta_v = -3\,\text{K}$ | $\Delta\Theta_v = -1\,\text{K}$ | $\Delta\Theta_v = +1\,\text{K}$ | $\Delta\Theta_v = +3\,\text{K}$ |
|---|---|---|---|---|
| 50 | 12.53 | 14.25 | 14.71 | 14.13 |
| 200 | 13.01 | 14.75 | 15.40 | 15.83 |
| 500 | 13.41 | 15.17 | 15.13 | 14.96 |
| 800 | 13.63 | 15.33 | 15.27 | 14.34 |
| 1500 | 13.90 | 15.55 | 15.60 | 13.74 |

**Table A3.** Temperature sensitivity (in $\% \, \text{K}^{-1}$) of melting for all $n_{\text{CCN}}$.

| $n_{\text{CCN}}$ (cm$^{-3}$) | $\Delta\Theta_v = -3\,\text{K}$ | $\Delta\Theta_v = -1\,\text{K}$ | $\Delta\Theta_v = +1\,\text{K}$ | $\Delta\Theta_v = +3\,\text{K}$ |
|---|---|---|---|---|
| 50 | $-3.91$ | $-6.18$ | $-8.93$ | $-9.00$ |
| 200 | $-3.45$ | $-4.85$ | $-6.86$ | $-7.81$ |
| 500 | $-3.73$ | $-5.11$ | $-6.36$ | $-7.22$ |
| 800 | $-3.83$ | $-5.38$ | $-6.39$ | $-7.07$ |
| 1500 | $-3.63$ | $-5.42$ | $-6.77$ | $-7.07$ |

**Table A4.** Temperature sensitivity (in $\% \, \text{K}^{-1}$) of rain evaporation for all $n_{\text{CCN}}$.

| $n_{\text{CCN}}$ (cm$^{-3}$) | $\Delta\Theta_v = -3\,\text{K}$ | $\Delta\Theta_v = -1\,\text{K}$ | $\Delta\Theta_v = +1\,\text{K}$ | $\Delta\Theta_v = +3\,\text{K}$ |
|---|---|---|---|---|
| 50 | 14.33 | 17.20 | 17.67 | 18.04 |
| 200 | 15.96 | 19.77 | 22.06 | 23.94 |
| 500 | 16.36 | 20.51 | 23.49 | 25.88 |
| 800 | 16.34 | 20.31 | 23.49 | 25.29 |
| 1500 | 16.06 | 19.72 | 22.92 | 23.18 |

**Table A5.** Temperature sensitivity (in $\% \, \text{K}^{-1}$) of total water vapour inflow $I$ for all $n_{\text{CCN}}$.

| $\Delta\Theta_v = -3\,\text{K}$ | $\Delta\Theta_v = -1\,\text{K}$ | $\Delta\Theta_v = +1\,\text{K}$ | $\Delta\Theta_v = +3\,\text{K}$ |
|---|---|---|---|
| 5.05 | 5.30 | 5.60 | 5.88 |





**Table A6.** Temperature sensitivity (in $\%\,\mathrm{K}^{-1}$) of condensation ratio $CR$ for all $n_{\mathrm{CCN}}$.

| $n_{\mathrm{CCN}}$ (cm$^{-3}$) | $\Delta\Theta_v = -3\,\mathrm{K}$ | $\Delta\Theta_v = -1\,\mathrm{K}$ | $\Delta\Theta_v = +1\,\mathrm{K}$ | $\Delta\Theta_v = +3\,\mathrm{K}$ |
|---|---|---|---|---|
| 50 | $-4.82$ | $-4.36$ | $-3.81$ | $-3.45$ |
| 200 | $-4.85$ | $-4.39$ | $-3.83$ | $-3.44$ |
| 500 | $-4.87$ | $-4.41$ | $-3.83$ | $-3.39$ |
| 800 | $-4.88$ | $-4.44$ | $-3.86$ | $-3.38$ |
| 1500 | $-4.89$ | $-4.50$ | $-3.92$ | $-3.39$ |

**Table A7.** Temperature sensitivity (in $\%\,\mathrm{K}^{-1}$) of precipitation efficiency $PE$ for all $n_{\mathrm{CCN}}$.

| $n_{\mathrm{CCN}}$ (cm$^{-3}$) | $\Delta\Theta_v = -3\,\mathrm{K}$ | $\Delta\Theta_v = -1\,\mathrm{K}$ | $\Delta\Theta_v = +1\,\mathrm{K}$ | $\Delta\Theta_v = +3\,\mathrm{K}$ |
|---|---|---|---|---|
| 50 | $-1.13$ | $-1.73$ | $-1.53$ | $-1.32$ |
| 200 | $-0.41$ | $-1.08$ | $-1.44$ | $-1.44$ |
| 500 | $0.19$ | $-0.44$ | $-1.16$ | $-1.76$ |
| 800 | $0.49$ | $-0.10$ | $-0.80$ | $-1.79$ |
| 1500 | $0.95$ | $0.42$ | $-0.21$ | $-1.54$ |

**Table A8.** Temperature sensitivity (in $\%\,\mathrm{K}^{-1}$) of drying ratio $DR$ for all $n_{\mathrm{CCN}}$.

| $n_{\mathrm{CCN}}$ (cm$^{-3}$) | $\Delta\Theta_v = -3\,\mathrm{K}$ | $\Delta\Theta_v = -1\,\mathrm{K}$ | $\Delta\Theta_v = +1\,\mathrm{K}$ | $\Delta\Theta_v = +3\,\mathrm{K}$ |
|---|---|---|---|---|
| 50 | $-6.12$ | $-6.16$ | $-5.28$ | $-4.63$ |
| 200 | $-5.32$ | $-5.52$ | $-5.22$ | $-4.74$ |
| 500 | $-4.65$ | $-4.88$ | $-4.95$ | $-4.97$ |
| 800 | $-4.31$ | $-4.54$ | $-4.63$ | $-4.98$ |
| 1500 | $-3.80$ | $-4.06$ | $-4.12$ | $-4.78$ |





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
