# Peer review of "Temperature and CCN sensitivity of orographic precipitation enhanced by a mixed-phase seeder-feeder mechanism"

_EGUsphere, 2022_

## Referee Comment (RC3)

**EGUSPHERE PAPER REVIEW**

**Temperature and Cloud Condensation Nuclei (CCN) sensitivity of orographic precipitation enhanced by a mixed-phase seeder-feeder mechanism**

By Thomas et al.

**General comments:**

Motivated by the rise in water vapour capacity of a globally warmer atmosphere, and the increasing frequency of extreme rainfall events, in this article, Thomas et al. evaluate the microphysical response of mid-latitude orographic rainfall to perturbations of temperature and CCN concentration. This study applies the use of "piggybacking" (Grabowski 2014) sensitivity experiments of rainfall from mixed-phase orographic clouds which, to the authors' and this reviewer's knowledge, is the first study to do so. This reviewer agrees that the piggybacking method is a robust technique to isolate the effects of warming and CCN concentrations on orographic precipitation. Interesting findings are presented in this article such as:

1.  Rainfall increase in a warming climate is significantly less than increase in precipitable water.
2.  A surface rain budget analysis reveals that the negative temperature sensitivity of the condensation ratio and the increase of sub-cloud rain evaporation dampen the rainfall enhancement in a warmer climate.
3.  Decreasing the CCN concentration speeds up the microphysical processing, esp. rain growth by collision-coalescence, which leads to an increase in total rainfall. This is consistent with previous model sensitivity studies, such as Chen et al. (2010), which have shown that decreasing CCN number in mixed-phase clouds results in fewer but larger cloud droplets, but also fewer ice crystals; its effect on surface precipitation depends on the interplay between the increased warm-rain production and the decreased or increased ice-phase precipitation. Precipitation responds nonlinearly to CCN number change, causing precipitation decrease in high CCN concentration environments but showing no clear tendency in low CCN concentration environments (Chen et al. 2010).
4.  In clean air (low CCN concentration) the sensitivity of rainfall to temperature is systematically smaller. In fact, the CCN and temperature sensitivities are to a large extent independent, and additive.

As is clearly stated in the section titled Limitations and Potential Solutions, this study lacks more robust conclusions that could be derived from a larger statistical ensemble of various orographic rainfall events, including different cloud depths, stability profiles, and wind profiles. Since this project was particularly based on a specific flood event in the UK (the Cumbria flood which occurred in December 2015), we suggest modifying the title to add ": a case study"

Overall, this paper deserves publication.

**Major Comments:**

1.  The statements on **Ln35** "where low mountain ranges would otherwise not efficiently produce precipitation", and on **Ln41** "nor do they have to be in the same thermodynamic state" miss a key point. Surely deeper clouds in which only warm rain processes operate experience orographic

enhancement, as low-level lifting enhances low-level cloud LWC. But that is not a seeder-feeder effect. The older literature was vague about this for lack of in situ observations, but the key to the seeder-feeder mechanisms as described in many more recent observational and modelling papers is *the lack of ice crystals in the shallow supercooled cloud layer*. In other words, the seeder-feeder mechanism is a mixed-phase process. See, for instance, Houze (2014, Cloud Dynamics 2$^{nd}$ Edition), p. 148-152. How do you define "cloud layer" in the statement on **Ln40** "The seeder-feeder mechanism does neither require the two cloud layers to be vertically separated"? You are correct if you refer to the water-saturated layers (those containing cloud droplets). But a defining aspect of the seeder-feeder mechanism is that falling ice particles (ice or snow in your model) reach the lower liquid-saturated layer, which is the case in your simulations (Fig. 6a and c). In other words, vertical continuity of ice particles is required. Please clarify/correct the text in the Introduction and elsewhere. For instance, in Section 5.1. And the last sentence of the Summary should be omitted.

2. Examination of the effect of a warmer climate on precipitation is done in an unconventional way. Why not uniformly change the Θv and qv (to maintain constant RH) at the lateral and surface boundaries? By changing the temperature only for the microphysics scheme (as mentioned on **Ln127**), your model is dynamically inconsistent. In doing so, you underestimate LW radiation from cloud base to the surface and from cloud top to outer space and evaporation from the sea or land surface, for instance. You are not representing a true warmer climate. The piggybacking technique is intended to separate microphysical and dynamical effects of changes in microphysical properties. Dynamical effects refer to changes in buoyancy or stability that might result in convection (for instance), altering the precipitation. But radiative and surface energy processes are not dynamical. Here, you treat the temperature change as a microphysical sensitivity (the atmosphere simply carries more water vapor and saturates at a higher q value). It is OK to call this a "piggybacking" method, although that is unusual. You claim that it removes the dynamical effects. It removes not just dynamical effects, also radiative, surface, PBL, and other effects. **I am asking that you do your model sensitivity analysis also the more conventional way, i.e. changing the Θv and qv in your driver dataset**. That will include all these other effects. This will quantify the importance of all these other effects on P. Given the strong winds and the small domain, I suspect the difference will be small, as stated also around Ln420.

3. **Eqn (7)** is misleading. The first term on the RHS (dyn) is incorporated in the other terms (CR and PE). All you can say here is that the T sensitivity can be broken down into three terms (not four). Then you can quantify these terms either using the "piggybacking" method, or the convectional method, as mentioned.

4. As discussed in the text and shown quite nicely (Fig. 4, Fig. 6c, Fig. 7), warming seems to shift the precip distribution from the central valley (Eden Valley?) to the upstream mountains (the peaks of the Lakes district). This shift in precip distribution explains the main conclusion of the paper, that the enhancement of rainfall due to warming is higher over the highest altitudes than over the entire domain. This is a key take-away in my opinion, and is dependent on the detailed terrain configuration. The results may be quite different for a different terrain layout. A more general treatise warrants evaluation using idealized terrain, as suggested already around Ln434. Maybe this has been done already, if so, please add a reference. If not, than here is a suggestion for a follow-up paper.

**Minor Comments:**

- **Ln 20:** the *relative* increase is stronger for lower temperatures. The absolute increase decreases with decreasing temperatures
- **Ln 29:** RH is assumed to *(be)\** constant... Refer to the CMIP6 ensemble mean or other reference, specifically for atmospheric rivers maybe. This is more than a hypothetical assumption, it is rooted in climate simulations under the synoptic conditions of interest, and that should be mentioned. On Ln 123, you refer to (Pörtner et al., 2022).
- **Ln 40:** The seeder-feeder mechanism *(thus)\** neither require …
- **Ln53:** Given the focus on extreme orographic precip, I suggest referring to the many studies of atmospheric rivers impacting coastal (or inland) terrain.
- **Ln57:** Preconditions → Upstream conditions (that is, they should persist during the storm)
- **Ln59:** strong wind, moist air: why not refer to IVT? Surely Browning did not use that quantity, but science has evolved.
- **Ln63:** last bullet: suggest simplifying this to: the terrain must be sufficiently high to lift the BL air mass above its LCL

**Fig. 2** Some corrections in the caption: Cross section used in (c), not (b). Also, what is "see 3"? Precip normally is expressed in depth (mm), rather than kg m$^{-2}$. The latter units may alienate some users. This change affects many Figs. The vertical velocity in (c) is quite coarse, it appears to be outer domain data, whereas the cross-section is entirely in the inner domain. At ~500 m grid resolution, I expect far more detail, including transient features. Also, can you please increase the figure size or plot size especially 2b. That plot should include the topo. In include an example here (screenshot from https://maps-for-free.com/), because I need it to interpret your subsequent figures.

[Figure]

- **Ln 68 & 69:** "both" applies to two traits, but there are three … suggest: to become more frequent and longer-lived, as well as more enriched with water vapour...

- **Ln 71:** pls quantify the integrated water vapor (PW) and integrated vapor transport (IVT), and infer that this event classifies as an atmospheric river.
- **Ln 97:** Use either (-4.2 ∘ to −1.8∘) or (4.2∘ E to 1.8 ∘ E) Longitude and from (53.5∘ N to 55.1∘ N) latitude. **NB: Longitudes are from West to East and Latitudes are from North to South.**
- **Ln 117**: Θv instead of absolute temperature is used *(to)\** preserve static stability.
- **Ln 120-121:** Initially, qv is *(adjusted)\** to the perturbed value of Θv...
- **Ln 167:** After the moist air *(enters)* LAKEDISTR (phase 1), it is forced to *(ascend)* **over** the mountain barrier.
- **Ln 373:** that can lead to an upwind shift *(in)* the distribution of precipitation.
- **Ln 380:** This discrepancy in αP may be due *(to)* the choice of the integration domain...
- **Ln 396:** the negative sensitivity of CR *(yields)* a total decrease in DR...

**Tables A1 and A2:** Can you account for the seemingly different temperature sensitivity trend for the PB-plus3-CCN? Sensitivity increases from just the nCCN = 50 cm$^{-3}$ to nCCN = 200 cm$^{-3}$. Temperature sensitivity of average rainwater content and accretion tend to decrease as the concentration of CCN increases. Why?

**Tables A3 - A8:** Is there an explanation as to why temperature sensitivity of melting stays the same for nCNN = 800 cm$^{-3}$ and nCNN = 1500 cm$^{-3}$ at PB-plus3-CNN condition? Why are there inconsistencies in temperature sensitivities under PB-minus1-CNN and PB-plus1-CNN conditions? Why are there inconsistencies in the temperature sensitivity trends for the remaining parameters in tables A4 – A8?

---

## Author Comment (AC1)

**Response to the reviewers**

We thank the reviewers for their critical assessment of our work. In the following we address their concerns point by point.
* * *
**Reviewer 1**

The manuscript discusses the microphysical response of orographic precipitation to perturbations of temperature and cloud condensation nuclei (CCN) concentration. The authors provide technically sound analysis. There are, however, some comments that need to be addressed.

**Specific comment 1.1** — The manuscript contains English writing problems and needs to be carefully edited.

**Reply**: We have fixed a number of grammatical mistakes.

**Specific comment 1.2** — The first sentence of the Abstract is irrelevant to the subject of the manuscript and is better to be replaced by another sentence.

**Reply**: We have changed the first sentence to: *"The formation of orographic precipitation in mixed-phase clouds depends on a complex interplay of processes."*

**Specific comment 1.3** — In the abstract and elsewhere, please replace "orographic rainfall" with "orographic precipitation"

**Reply**: Done.

**Specific comment 1.4** — It is better to remove the following sentence in the abstract or move it to the Introduction Section to have enough space to discuss the obtained results: "The study is motivated by the increased water vapour capacity of the atmosphere in a warming climate and the increasing frequency of extreme rainfall events."

**Reply**: We have moved the sentence to the first paragraph of the introduction and reformulated it.

**Specific comment 1.5** — Line 5: write out the full words for "ICON"

**Reply**: Done.

**Specific comment 1.6** — The Introduction is much weaker than the other parts of the manuscript and could be substantially improved. For example, the first paragraph of the Introduction does not provide any important information about the topic of the manuscript. It is irrelevant and can be omitted.

**Reply**: We have changed the first paragraph as follows: *"Orographically enhanced severe precipitation events are impacted by the increased water vapour capacity of the atmosphere in a warming climate and the general trend of increasing frequencies of extreme precipitation events (Pörtner et al., 2022).*

*A detailed understanding of cloud microphysical processes that cause orographic precipitation is crucial to assess flood risk in the vicinity of low mountain ranges (Houze, 2012), now and in the future."*

Further changes in the introduction are outlined below.

**Specific comment 1.7** — Another example is in Line 22 where appropriate references and more clarification are required for the dynamic, thermodynamic, and microphysical contributions to precipitation changes under global warming. The microphysical contribution to precipitation changes is well discussed, but at least a few sentences can be added to explain the thermodynamic and dynamic contributions, particularly because the thermodynamic contribution to precipitation change is the main subject of the manuscript. As discussed in http://dx.doi.org/10.1007/s10584-022-03316-z, the precipitation response to climate change is regulated by two basic mechanisms, which include the wet-get-wetter mechanism and the warmer-get-wetter, both of which are referred to as the thermodynamic mechanism, while circulation changes under the impact of global warming which lead to precipitation changes are referred to as the dynamic change of precipitation.

**Reply**: We agree that a more elaborate introduction to previous literature on the different contributions to precipitation changes was missing in our manuscript. We have reformulated the entire paragraph and included several references to original works on the CC scaling, global increase of water vapour and precipitation, as well as the thermodynamic effect. Since piggybacking suppresses any change in the dynamics, discussing the dynamical effect is beyond the scope of our manuscript.

*"The relationship that dictates the atmospheric water vapour content is the Clausius-Clapeyron (CC) equation for saturation vapour pressure of water $e_{sat}$. In the atmospheric temperature range of interest in this work, $e_{sat}$ increases between $6.5 \% \, \mathrm{K}^{-1}$ and $9 \% \, \mathrm{K}^{-1}$. As shown in Fig. 1, the relative increase is stronger for lower temperatures, i.e. higher up in the atmosphere. The value of $dlog(e_{sat})/dT$ for a given temperature is referred to as CC scaling throughout the paper. To a first approximation, relative humidity is constant in a warmer climate, since enhanced evaporation balances the increased capacity of the atmosphere to hold water vapour (Held and Soden, 2006). In particular, this holds for the upstream conditions of coastal orographic precipitation (Payne et al., 2020). A naive assumption might be that total precipitation increases by the same rate as atmospheric water vapour, but climate models predict that the global precipitation increases more slowly with global mean temperature than CC scaling (Allen and Ingram, 2002). Deviations from this assumption can be due to changes in dynamics (Pfahl et al., 2017), in the rate of condensation (thermodynamic effect) or due to changes in the microphysical processing of cloud water (O'Gorman, 2015). The thermodynamic effect originates in the temperature sensitivity of the CC scaling. Regionally, this leads to wet regions getting wetter, dry regions getting dryer (Held and Soden, 2006). Locally, the relative increase in condensation is larger at high altitudes, according to the relative increase in water vapour. Siler and Roe (2014) showed that due to the temperature sensitivity of the moist adiabatic lapse rate, the increase in condensation is damped. It can lower the CC scaling by up to $4 \% \, \mathrm{K}^{-1}$ (Siler and Roe, 2014). Deviations from the CC scaling caused by cloud microphysics, which are characterized by the precipitation efficiency (PE), are the subject of this work."*

**Specific comment 1.8** — Line 29: replace "by constant" with "be constant"

**Reply**: Done.

**Specific comment 1.9** — Line 34: Replace "English West coast" with "British West coast"

**Reply**: Done.

**Specific comment 1.10** — Line 45-46. Total precipitation might reduce or slightly change under a higher CCN concentration, but please note that some evidence suggests that heavy precipitation might increase in a polluted environment. For example, see the following paper: http://dx.doi.org/10.1016/j.atmosres.2016.10.021. This needs to be discussed in the Introduction, particularly because the importance of extreme precipitation and the risk of flooding is emphasized in the first paragraph of the introduction.

**Reply**: We have added the following sentence in line 49: *"This can lead to complex responses of the distribution of precipitation. E.g., Alizadeh-Choobari and Gharaylou (2017) found that for a case of convectively enhanced frontal precipitation over a mountainous region, light precipitation was reduced but moderate and strong precipitation intensified when the concentration of hygroscopic aerosols was increased."*

**Specific comment 1.11** — Line 49-50: This sentence does not mean anything: Which effect dominates highly depends on the synoptic conditions (e.g. whether convection is involved) and on the mountain geometry.

**Reply**: We slightly reformulate the sentence and have decided to keep it here, since we think that the variability of the net effect of aerosol on orographic precipitation is important for the following discussion. *"In general, the net effect highly depends on the synoptic conditions (e.g. whether convection is involved), on the model's treatment of aerosols and on the mountain geometry (Kunz and Kottmeier, 2006)."*

**Specific comment 1.12** — Line 56: westward or eastward?

**Reply**: Thanks for pointing out this error. We have corrected the sentence to *"At the British West coast, moist air moving eastward over the Atlantic is lifted by low mountain ranges and produces orographic precipitation over and downstream of these coastal mountains."*.

**Specific comment 1.13** — Line 93: write out the full words in the first use of "DWD"

**Reply**: Done.

**Specific comment 1.14** — Parameterization schemes that are used for the boundary layer, radiation, etc. should be mentioned in Section 2.1

**Reply**: The turbulence and shallow convection parameterizations were already mentioned in the text. We have added mention of the radiation scheme (Rapid Radiative Transfer Model RRTM, Mlawer et al., 1997).

**Specific comment 1.15** — Lines 152-155: Rephrase the sentence.

**Reply**: We have rephrased the sentence as follows: *"The black dashed contours in Fig. 2 outline the evaluation domain, which is aligned with the mean wind speed and is referred to as LAKEDISTR. Figure 3 shows the simulated hourly rates of surface rainfall, averaged over LAKEDISTR. The continuous*

*line marks the 16 h period used to calculate all totals defined in the following section. Starting on 5 December 2015, 10:00 UTC and ending on 6 December 2015, 02:00 UTC, it is chosen such that it includes the highest rainfall rates and ends before a rapid decrease in rainfall is observed."*

**Specific comment 1.16** — Fig.4: It is better to start the label bar with white color representing zero value. Instead of using kg m-2 for the unit of precipitation, please consider writing it as mm for this figure and the others, although the values do not change.

**Reply**: We have replaced kg m$^{-2}$ by mm in Figure 2, 3, 4 and 5. The colour bar was chosen to start with light yellow colour, representing values of surface precipitation between 1 and 50 mm. There are no regions without any precipitation during the evaluation time span.

**Specific comment 1.17** — Line 198-201. It is expected that heavy precipitation experiences the highest increase in response to warming, while light precipitation experiences a slight change or even decrease. It is interesting to compare the obtained results with those of http://dx.doi.org/10.1002/met.1724. This paper is already cited in the Introduction, but here or in the discussion section you can compare your results with their results and discuss the reasons for such behavior. In this paper, it is noted that more hygroscopic aerosols in the atmosphere could potentially increase heavy precipitation but reduce light precipitation. The author argued that an ample influx of water vapour over the regions with heavy precipitation could contribute to an increase of heavy precipitation under a polluted atmosphere, while less efficient autoconversion processes and/or increased cloud-top evaporation may contribute to a decrease in light precipitation. Could the same reasons contribute to an increase in heavy precipitation and a decrease in light precipitation in response to warming in your study? Please clarify. Such discussions could improve the quality of the manuscript.

**Reply**: Our analysis has shown that the majority of the rain mass is formed through mixed-phase processes (see Fig. 6), while the arguments of Alizadeh-Choobari (2018) relate to formation of cloud droplets by activation of hygroscopic aerosols and their growth in the liquid phase. In our case, it is the vertical alignment of the melting and accretion processes which dominate the rain enhancement in the high precipitation regions. However, we have not investigated the horizontal distribution of rainfall in the different CCN scenarios. This could be a topic for future studies. We have added a sentence in the discussion part: *"Furthermore, a similar shift in the spatial distribution of precipitation, but related to an increase in CCN and enhanced condensation, was found by (Alizadeh-Choobari, 2018). A similarity between these effects is that precipitation was found to increase most in regions where the conversion of inflowing water vapour to rain is most efficient."*

**Specific comment 1.18** — Line 282: remove "a" before "only"

**Reply**: Done.

**Specific comment 1.19** — Line 407: Replace "in climate change" with "under climate change"

**Reply**: Done.

**Reviewer 2**

This manuscript describes be sensitivity of precipitation efficiency with increased temperature and CCN concentration. I find the manuscript mostly well written and organized. The conclusions are justified and I think this manuscript should be published after some minor revisions.

**Specific comment 2.1** — Page 1, line 21: Can you provide a citation for the statement: "If there were no changes in cloud dynamics and microphysics, the total precipitation would increase by the same rate as total vapour inflow"

**Reply**: We changed the sentence as follows: *"A naive assumption would be that total precipitation increases by the same rate as atmospheric water vapour."*

**Specific comment 2.2** — Page 2, line 35: When I first read through the manuscript, I questioned why specifically the English West Coast was mentioned. Later, when learning that the location of the Cumbria flood is at the English Coast, I realized the importance. However, the Cumbria flood has so far only been mentioned in the Abstract (and not given a more general location (the English West Coast)). I think a description of the Cumbria flood needs to come before the statement of "enhanced precipitation along the English West coast", or this statement needs to be moved.

**Reply**: We have removed this part of the sentence.

**Specific comment 2.3** — Page 3, line 56: I suggest rewording to: "At the English West Coast, moist air flows that is moving westward over the Atlantic are . . . ."

**Reply**: We have changed the sentence as follows: *"At the British West coast, moist air moving eastward over the Atlantic is lifted by low mountain ranges and produces orographic precipitation over and downstream of these coastal mountains."*

**Specific comment 2.4** — Page 3, line 64: I suggest rephrasing to: "Only then are low mountain ranges (up to 1 km height) sufficient to ..."

**Reply**: Done.

**Specific comment 2.5** — Page 3, line 71: How long did the flood last? From 5th December to?

**Reply**: We have added: *"The Cumbria flood from 5th to 6th December 2015..."*

**Specific comment 2.6** — Page 3, line 71 to 77: This is a nice description of the Cumbria Flood and its area. I finally understand why the English West Coast is mentioned earlier. As mentioned in comment above, some description of the Cumbria Flood (at least location) needs to come before describing precipitation on the English West Coast so the reader understands why there is a focus on the English West coast.

**Reply**: We removed the part containing "English West Coast" in line 34. Now, both the words "British" and "Cumbria" first appear in line 51 onwards, where the case is described more closely.

**Specific comment 2.7** — Page 5, line 96: The triangular cells. This is the first time triangular cells are mentioned. Perhaps add a sentence how the grids are used in ICON.

**Reply**: We have added *"The model works on an icosahedral grid to provide a nearly homogeneous coverage of the globe."* in line 93.

**Specific comment 2.8** — Page 5, lines 112-114: On line 112 it is mentioned four sets of all microphysical prognostic variables, while on line 114, it is said "the five microphysical variable sets". Is it four or five or am I mistaken and there are different microphysical variable sets?

**Reply**: We have clarified as follows: *"The cloud microphysics scheme is called five times per time step, once for the reference simulation and once for each of the four piggybacking sets. Thus, at each time step, all of the five microphysical variable sets are updated. Therefore, each simulation results in five complete output variable sets with different microphysics but identical wind (see Fig. 2a and c) and pressure fields."*

**Specific comment 2.9** — Page 5, Line 116: Not sure why Fig2b is referenced.

**Reply**: We have substituted this by "Fig 2a and c".

**Specific comment 2.10** — Page 5, line 117: suggest to add "to" : "... temperature is used to preserve static ..."

**Reply**: Done.

**Specific comment 2.11** — Page 6, line 152: Are the hourly rates of rainfall from observations or the model?

**Reply**: They are from the model. We have added *"simulated"*.

**Specific comment 2.12** — Page 14, line 282: Remove "a" before "only"

**Reply**: Done.

**Specific comment 2.13** — Page 16: Line 342: What is the reason for that the temperature sensitivity is higher in more polluted regions? This might have been stated earlier, but it would be good to spell it out again.

**Reply**: This is due to smaller drop sizes as well as a different partitioning of warm- and cold phase processes and a resulting less negative sensitivity of PE (see Fig. 9 and Table A7). This is also discussed at the end of section 4.2. Nevertheless, we have added this information here again. To keep the manuscript concise, we would like to avoid repeating the detailed analyses for the different CCN scenarios.

**Specific comment 2.14** — Page 18. Lines 394 – 401. I think it is worth including the study by Eidhammer et al (2018) here in the discussion as well. They looked at changes in drying ratio in a changing climate. They noted that the shape of the mountains and the wind speeds also have impacts on how the drying ratio changes with temperature. The difference between the current study and Siler and Roe and Kirshbaum and Smith can therefore also be additional attributed to

differences in the orography and windspeeds. Typically, wider mountain ranges have a lower change in drying ratio due to temperature increases compared to narrower mountains.

**Reply**: Thanks for the suggestion. We have added after line 401: *"Other possible reasons for different $\alpha_{DR}$ values are mountain width and horizontal wind speed. Eidhammer et al. (2018) found that the decrease of $\alpha_{DR}$ is stronger for wider mountains, because the microphysical timescale is larger. Furthermore, in the case of narrow mountains (width less than $50$ km), the decrease of $\alpha_{DR}$ is lower for lower horizontal wind speed. In fact, although absolute $\alpha_{DR}$ is lower in Kirshbaum and Smith (2008), the half width in their experiment is more than three times bigger than in this study. Horizontal wind speed in Kirshbaum and Smith (2008) and Siler and Roe (2014) is comparable to this study. Moreover, Siler and Roe (2014) found little dependence of $\alpha_{DR}$ on horizontal wind speed."*

**Specific comment 2.15** — Page 19, line 416. Eidhammer et al (2018) used downscaled climate simulation for their study on DR. In that study, the horizontal velocity changed between the current and future climate scenarios, and they showed the impact the changed velocity field had on the DR.
(Eidhammer et al. 2018: Winter precipitation efficiency of mountain ranges in the Colorado Rockies under climate change, J. Geophys. Res. 123, 2573-2590, https://doi.org/10.1002/2017JD027995.)

**Reply**: We have added the following sentences: *"Eidhammer et al. (2018) used the pseudo-global warming technique to simulate a future climate scenario. Their method also keeps the large-scale dynamics unchanged but allows vertical velocities to adjust.*

**Specific comment 2.16** — Page 19, line 434. I suggest adding citation of Kirshbaum et al and Eidhammer et al here regarding the rescaling of orography.

**Reply**: Done.
* * *
**Reviewer 3**

Motivated by the rise in water vapour capacity of a globally warmer atmosphere, and the increasing frequency of extreme rainfall events, in this article, Thomas et al. evaluate the microphysical response of mid-latitude orographic rainfall to perturbations of temperature and CCN concentration. This study applies the use of "piggybacking" (Grabowski 2014) sensitivity experiments of rainfall from mixed-phase orographic clouds which, to the authors' and this reviewer's knowledge, is the first study to do so. This reviewer agrees that the piggybacking method is a robust technique to isolate the effects of warming and CCN concentrations on orographic precipitation. Interesting findings are presented in this article such as: 1. Rainfall increase in a warming climate is significantly less than increase in precipitable water. 2. A surface rain budget analysis reveals that the negative temperature sensitivity of the condensation ratio and the increase of sub-cloud rain evaporation dampen the rainfall enhancement in a warmer climate. 3. Decreasing the CCN concentration speeds up the microphysical processing, esp. rain growth by collision-coalescence, which leads to an increase in total rainfall. This is consistent with previous model sensitivity studies, such as Chen et al. (2010), which have shown that decreasing CCN number in mixed-phase clouds results in fewer but larger cloud droplets, but also fewer ice crystals; its effect on surface precipitation depends on the interplay between the increased warm-rain production and the decreased or increased

ice-phase precipitation. Precipitation responds nonlinearly to CCN number change, causing precipitation decrease in high CCN concentration environments but showing no clear tendency in low CCN concentration environments (Chen et al. 2010). 4. In clean air (low CCN concentration) the sensitivity of rainfall to temperature is systematically smaller. In fact, the CCN and temperature sensitivities are to a large extent independent, and additive.

Overall, this paper deserves publication.

Since this project was particularly based on a specific flood event in the UK (the Cumbria flood which occurred in December 2015), we suggest modifying the title to add ": a case study"

**Reply**:  We agree and have added *": a case study for the 2015 Cumbria flood"* to the title.

**General comment 3.1**  —  The statements on Ln35 "where low mountain ranges would otherwise not efficiently produce precipitation", and on Ln41 "nor do they have to be in the same thermodynamic state" miss a key point. Surely deeper clouds in which only warm rain processes operate experience orographic enhancement, as low-level lifting enhances low-level cloud LWC. But that is not a seeder-feeder effect. The older literature was vague about this for lack of in situ observations, but the key to the seeder-feeder mechanisms as described in many more recent observational and modelling papers is the lack of ice crystals in the shallow supercooled cloud layer. In other words, the seeder-feeder mechanism is a mixed-phase process. See, for instance, Houze (2014, Cloud Dynamics 2nd Edition), p. 148-152. How do you define "cloud layer" in the statement on Ln40 "The seeder-feeder mechanism does neither require the two cloud layers to be vertically separated"? You are correct if you refer to the water-saturated layers (those containing cloud droplets). But a defining aspect of the seeder-feeder mechanism is that falling ice particles (ice or snow in your model) reach the lower liquid-saturated layer, which is the case in your simulations (Fig. 6a and c). In other words, vertical continuity of ice particles is required. Please clarify/correct the text in the Introduction and elsewhere. For instance, in Section 5.1. And the last sentence of the Summary should be omitted.

**Reply**:  We disagree with the reviewer's opinion that a seeder-feeder effect can only occur if the lower cloud layer is supercooled. In our case, it is not - a larger part of the liquid cloud water is located at temperatures above $0°$C (see Fig. 6c), and the rainfall enhancement occurs through accretion of cloud droplets onto rain generated from melting of ice/graupel particles falling from the upper cloud. Therefore, the mixed-phase nature of the cloud system is crucial, although the rain enhancement i's a liquid phase process in our case. This is in line with the definition of the seeder-feeder process given in the AMS's Glossary of Meteorology (https://glossary.ametsoc.org/wiki/Seeder-feeder). We have therefore removed the sentence on the separation (or not) of the cloud layers. We have kept the description in Section 5.1 as we think it is correct. The last sentence of the summary has been reformulated to *"This implies that severe rainfall in mountainous regions via the seeder-feeder mechanism may increase in future."* In our opinion, this is an important implication of our findings and should not be omitted. However, we have now included the caveat *"If our findings are transferable to similar cases, ..."* two sentences earlier.

**General comment 3.2**  —  Examination of the effect of a warmer climate on precipitation is done in an unconventional way. Why not uniformly change the $\Theta_v$ and qv (to maintain constant RH) at the lateral and surface boundaries? By changing the temperature only for the microphysics scheme (as mentioned on Ln127), your model is dynamically inconsistent. In doing so, you underestimate

LW radiation from cloud base to the surface and from cloud top to outer space and evaporation from the sea or land surface, for instance. You are not representing a true warmer climate. The piggybacking technique is intended to separate microphysical and dynamical effects of changes in microphysical properties. Dynamical effects refer to changes in buoyancy or stability that might result in convection (for instance), altering the precipitation. But radiative and surface energy processes are not dynamical. Here, you treat the temperature change as a microphysical sensitivity (the atmosphere simply carries more water vapor and saturates at a higher q value). It is OK to call this a "piggybacking" method, although that is unusual. You claim that it removes the dynamical effects. It removes not just dynamical effects, also radiative, surface, PBL, and other effects. I am asking that you do your model sensitivity analysis also the more conventional way, i.e. changing the $\Theta_v$ and qv in your driver dataset. That will include all these other effects. This will quantify the importance of all these other effects on P. Given the strong winds and the small domain, I suspect the difference will be small, as stated also around Ln420.

**Reply**: The reviewer is correct that we have extended the use of "piggybacking" to a new kind of perturbation, namely temperature. We agree that this not only suppresses dynamical feedbacks, but also feedbacks on radiation and turbulence. To clarify this, we have modified several passages in the manuscript. The last sentence at the end of the introduction now reads: *"This method allows to evaluate microphysical sensitivities without the potentially confusing impacts of changes in the dynamics, radiation and other processes."* and the beginning of section 2.2 was modified as follows: *"Piggybacking is a simple and computationally efficient method to separate microphysical sensitivities from feedbacks to dynamics, radiation and other processes (Grabowski, 2014). It is motivated by a challenge that all sensitivity studies with fully interactive models encounter: perturbations of microphysical parameters cause feedbacks on the thermodynamic state of the atmosphere (e.g. on temperature and buoyancy by latent heating) and consequently on the dynamics of the system, i.e. wind, pressure and static stability, as well as radiative fluxes and turbulence. Here we use piggybacking (a) to quantify the immediate microphysical sensitivity of orographic rainfall to changes of thermodynamic conditions (here: changes in temperature) and (b) to investigate the response to changes in microphysical parameters, specifically the CCN number concentration."* The isolation of the temperature effect on microphysics was the purpose of this exercise, therefore we do not see it as a weakness. Full pseudo-climate warming experiments would have required further choices e.g. on adjustment of the surface temperature, surface fluxes, stability of the upper troposphere above the clouds, etc. These simulations would have required a longer spin-up and would have been difficult to compare directly to the piggybacked experiments. Therefore we have not conducted these experiments. However, for the CCN sensitivity study, we have compared simulations with piggybacked CCN and simulations with CCN changed in the driving simulation. The results are shown below as the deviation in 24-h average rainfall between the reference simulation and simulations with different CCN concentrations in the driving and piggybacked simulation (Thomas, 2022). Comparing the vertical and horizontal changes, the dynamical and other contributions that we removed by using piggybacking are significantly smaller than the microphysical contribution.

[Figure]

**General comment 3.3** — Eqn (7) is misleading. The first term on the RHS (dyn) is incorporated in the other terms (CR and PE). All you can say here is that the T sensitivity can be broken down into three terms (not four). Then you can quantify these terms either using the "piggybacking" method, or the convectional method, as mentioned.

**Reply**: It is correct that $I$, $PE$ and $CR$ can also change by pertubations of the dynamics, but here we incorporate all those changes into the first term on the RHS of the first line of Eqn (7). According to Eqn (2) we assume that the dynamical and the microphysical sensitivities are independent and additive. If piggybacking is applied, the first term vanishes (third line). For clarification, we have added the following explanation: *"Here, all dynamical feedbacks are summarized in $\alpha_{P,dyn}$, while $\alpha_I$, $\alpha_{PR}$ and $\alpha_{PE}$ on contain the microphysical and thermodynamic contributions."*

**General comment 3.4** — As discussed in the text and shown quite nicely (Fig. 4, Fig. 6c, Fig. 7), warming seems to shift the precip distribution from the central valley (Eden Valley?) to the upstream mountains (the peaks of the Lakes district). This shift in precip distribution explains the main conclusion of the paper, that the enhancement of rainfall due to warming is higher over the highest altitudes than over the entire domain. This is a key take-away in my opinion, and is dependent on the detailed terrain configuration. The results may be quite different for a different terrain layout. A more general treatise warrants evaluation using idealized terrain, as suggested already around Ln434. Maybe this has been done already, if so, please add a reference. If not, than here is a suggestion for a follow-up paper.

**Reply**: While several previous studies have used idealised orography (Siler and Roe, 2014; Muhlbauer and Lohmann, 2008), we are not aware of any systematic investigation of CCN effects over orography with more than one peak. So this is indeed a useful direction for future studies. We notice that other studies found different changes of rain distribution with increasing temperature or CCN. As discussed in section 5.1, Siler and Roe 2014 found the opposite effect in their idealized case study. Since the thermodynamic effect (higher increase of condensation at high altitudes) leads to a downwind shift, whereas the faster microphysical processing leads to an upwind shift, these two effects always compete each other. In Ln 381ff., where the reason for the differences are discussed, we have added mention of the terrain structure

**Specific comment 3.1** — Ln 20: the relative increase is stronger for lower temperatures. The absolute increase decreases with decreasing temperatures.

**Reply**: We have added "relative".

**Specific comment 3.2** — Ln 29: RH is assumed to (be)* constant... Refer to the CMIP6 ensemble mean or other reference, specifically for atmospheric rivers maybe. This is more than a hypothetical assumption, it is rooted in climate simulations under the synoptic conditions of interest, and that should be mentioned. On Ln 123, you refer to (Pörtner et al., 2022).

**Reply**: Thanks for this suggestion. We have modified this sentence as follows: *"To a first approximation, relative humidity is constant in a warmer climate, since enhanced evaporation balances the increased capacity of the atmosphere to hold water vapour (Held and Soden, 2006). In particular, this holds for the upstream conditions of coastal orographic precipitation (Payne et al., 2020)."*

**Specific comment 3.3** — Ln 40: The seeder-feeder mechanism (thus)* neither require ...

**Reply**: As this section was reformulated, this sentence was removed.

**Specific comment 3.4** — Ln53: Given the focus on extreme orographic precip, I suggest referring to the many studies of atmospheric rivers impacting coastal (or inland) terrain.

**Reply**: Thanks for the suggestion. We refer to pertinent references on atmospheric rivers (Lavers and Villarini, 2015; Matthews et al., 2018) a little later in the text.

**Specific comment 3.5** — Ln57: Preconditions → Upstream conditions (that is, they should persist during the storm)

**Reply**: We have changed *preconditions* to *suitable conditions*, as not only upstream conditions are included in the list (e.g. terrain height).

**Specific comment 3.6** — Ln59: strong wind, moist air: why not refer to IVT? Surely Browning did not use that quantity, but science has evolved.

**Reply**: Thanks for the suggestion. We have changed the first bullet point to *"prevailing fast low-level winds advecting a moist air mass (Browning et al., 1975), i.e. high values of integrated water vapour transport (IVT)"*

**Specific comment 3.7** — Ln63: last bullet: suggest simplifying this to: the terrain must be sufficiently high to lift the BL air mass above its LCL

**Reply**: Done.

**Specific comment 3.8** — Fig. 2 Some corrections in the caption: Cross section used in (c), not (b). Also, what is "see 3"? Precip normally is expressed in depth (mm), rather than kg m-2. The latter units may alienate some users. This change affects many Figs.

**Reply**: We changed (b) to (d). We changed "see ..." to "period indicated in Fig. 3". We changed units in all affected figures to "mm".

**Specific comment 3.9** — The vertical velocity in (c) is quite coarse, it appears to be outer domain data, whereas the cross-section is entirely in the inner domain. At 500 m grid resolution, I expect far more detail, including transient features. Also, can you please increase the figure size or plot size especially 2b. That plot should include the topo. In include an example here (screenshot from https://maps-for-free.com/), because I need it to interpret your subsequent figures.

**Reply**: For Fig. 2d, the inner domain data has been interpolated to 200 evenly spaced points (approximately 800 m resolution) along the cross section. This information has been added to the caption. We have increased the size of Fig. 2. Furthermore, we have added a topographical map of the Lake District area (Fig. 2c).

**Specific comment 3.10** — Ln 68 & 69: "both" applies to two traits, but there are three ... suggest: to become more frequent and longer-lived, as well as more enriched with water vapour...

**Reply**: Done.

**Specific comment 3.11** — Ln 71: pls quantify the integrated water vapor (PW) and integrated vapor transport (IVT), and infer that this event classifies as an atmospheric river.

**Reply**: We have added the following: *"Storm Desmond, which caused the Cumbria flood in 2015, was accompanied by an atmospheric river (Lavers et al., 2016) with peak 24 hour mean IVT of more than 1100 $\mathrm{kg\,m^{-1}s^{-1}}$, the highest observed for an atmospheric river impacting the British Isles since at least 1979 (Matthews et al., 2018)."*

**Specific comment 3.12** — Ln 97: Use either (-4.2 ∘ to -1.8∘) or (4.2∘ E to 1.8 ∘ E) Longitude and from (53.5∘ N to 55.1∘ N) latitude. NB: Longitudes are from West to East and Latitudes are from North to South.

**Reply**: Corrected as follows: *"The inner nest extends from 4.2°W to 1.8°W longitude ... and from 53.5°N to 55.1°N latitude ..."*

**Specific comment 3.13** — Ln 117: $\Theta_v$ instead of absolute temperature is used (to)* preserve static stability.

**Reply**: Corrected.

**Specific comment 3.14** — Ln 120-121: Initially, qv is (adjusted)* to the perturbed value of $\Theta_v$...

**Reply**: Corrected.

**Specific comment 3.15** — Ln 167: After the moist air (enters) LAKEDISTR (phase 1), it is forced to (ascend) over the mountain barrier.

**Reply**: Corrected.

**Specific comment 3.16** — Ln 373: that can lead to an upwind shift (in) the distribution of precipitation.

**Reply**: Corrected.

**Specific comment 3.17** — Ln 380: This discrepancy in $\alpha_P$ may be due (to) the choice of the integration domain...

**Reply**: Corrected.

**Specific comment 3.18** — Ln 396: the negative sensitivity of CR (yields) a total decrease in DR...

**Reply**: Corrected.

**Specific comment 3.19** — Tables A1 and A2: Can you account for the seemingly different temperature sensitivity trend for the PB-plus3-CCN? Sensitivity increases from just the nCCN = 50 cm-3 to nCCN = 200 cm-3. Temperature sensitivity of average rainwater content and accretion tend to decrease as the concentration of CCN increases. Why?

**Reply**: We can only speculate here. The reduced temperature sensitivity of RWC for +3K could be due to the competition between increased melting (a source of rain) and increased evaporation (a sink of rain). At +3K, the layer in which these two processes can occur has the strongest vertical extent. Regarding the sensitivity to $n_{CCN}$ (in each column), it is interesting to note that it constantly decreases from the coldest to the warmest scenario, switching sign between +1K and +3K. This might be because the CCN sensitivity is a competition between warm and cold phase processes, which do not change consistently.

**Specific comment 3.20** — Tables A3 - A8: Is there an explanation as to why temperature sensitivity of melting stays the same for nCNN = 800 cm-3 and nCNN = 1500 cm-3 at PB-plus3-CNN condition? Why are there inconsistencies in temperature sensitivities under PB-minus1-CNN and PB-plus1-CNN conditions? Why are there inconsistencies in the temperature sensitivity trends for the remaining parameters in tables A4 – A8?

**Reply**: It is important to note that these numbers are not the result of a chain of processes occurring in one vertical column, but are averaged over three-dimensional time-varying fields. As can be seen in Fig. 6 for the temperature sensitivity and in the below figure for the CCN sensitivity, the processes and column water paths do not vary consistently at all locations along the mean flow. The integration domain and the evaluated time period are thus expected to have an impact on the results. Therefore we are not surprised by the apparent inconsistencies in the values in Tables A1 to A8. Nevertheless, the purpose of these tables is to put the numbers given in the summary section, which only refer to the changes for reference simulation, into context.

[Figure]

**References**

Alizadeh-Choobari, O.: Impact of aerosol number concentration on precipitation under different precipitation rates, Meteorological Applications, 25, 596–605, https://doi.org/10.1002/met.1724, 2018.

Alizadeh-Choobari, O. and Gharaylou, M.: Aerosol impacts on radiative and microphysical properties of clouds and precipitation formation, Atmospheric Research, 185, 53–64, https://doi.org/10.1016/j.atmosres.2016.10.021, 2017.

Allen, R. and Ingram, W.: Constraints on future changes in climate and the hydrologic cycle, Nature, 419, 224—-232, https://doi.org/10.1038/nature01092, 2002.

Browning, K., Pardoe, C., and Hill, F.: The nature of orographic rain at wintertime cold fronts, Quarterly Journal of the Royal Meteorological Society, 101, 333–352, https://doi.org/10.1002/qj.49710142815, 1975.

Eidhammer, T., Grubišić, V., Rasmussen, R., and Ikdea, K.: Winter precipitation efficiency of

mountain ranges in the Colorado Rockies under climate change, Journal of Geophysical Research: Atmospheres, 123, 2573–2590, 2018.

Grabowski, W. W.: Extracting microphysical impacts in large-eddy simulations of shallow convection, Journal of the Atmospheric Sciences, 71, 4493–4499, https://doi.org/10.1175/JAS-D-14-0231.1, 2014.

Held, I. M. and Soden, B. J.: Robust Responses of the Hydrological Cycle to Global Warming, Journal of Climate, 19, 5686 – 5699, https://doi.org/10.1175/JCLI3990.1, 2006.

Houze, R. A. J.: Orographic effects on precipitating clouds, Reviews of Geophysics, 50, RG1001, https://doi.org/10.1029/2011RG000365, 2012.

Kirshbaum, D. J. and Smith, R. B.: Temperature and moist-stability effects on midlatitude orographic precipitation, Q. J. R. Meteorological Society, 134, 1183–1199, https://doi.org/10.1002/qj.274, 2008.

Kunz, M. and Kottmeier, C.: Orographic enhancement of precipitation over low mountain ranges. Part I: Model Formulation and Idealized Simulations, Journal of Applied Meteorology and Climatology, 45, 1025–1040, https://doi.org/10.1175/JAM2390.1, 2006.

Lavers, D. A. and Villarini, G.: The contribution of atmospheric rivers to precipitation in Europe and the United States, Journal of Hydrology, 522, 382–390, https://doi.org/10.1016/j.jhydrol.2014.12.010, 2015.

Lavers, D. A., Pappenberger, F., Richardson, D. S., and Zsoter, E.: ECMWF Extreme Forecast Index for water vapor transport: A forecast tool for atmospheric rivers and extreme precipitation, Geophysical Research Letters, 43, https://doi.org/10.1002/2016gl071320, 2016.

Matthews, T. et al.: Super Storm Desmond: a process-based assessment, Environmental Research Letters, 13, 014024, https://doi.org/10.1088/1748-9326/aa98c8, 2018.

Muhlbauer, A. and Lohmann, U.: Sensitivity Studies of the Role of Aerosols in Warm-Phase Orographic Precipitation in Different Dynamical Flow Regimes, Journal of The Atmospheric Sciences - J ATMOS SCI, 65, 2522–2542, https://doi.org/10.1175/2007JAS2492.1, 2008.

O'Gorman, P. A.: Precipitation extremes under climate change, Current Climate Change Reports, 1, 49–59, https://doi.org/10.1007/s40641-015-0009-3, 2015.

Payne, A. E. et al.: Responses and impacts of atmospheric rivers to climate change, Nature Reviews Earth & Environment, 1, 143–157, https://doi.org/10.1038/s43017-020-0030-5, 2020.

Pfahl, S., O'Gorman, P. A., and Fischer, E. M.: Understanding the regional pattern of projected future changes in extreme precipitation, Nature Climate Change, 7, 423–427, https://doi.org/10.1038/nclimate3287, 2017.

Pörtner, H. O., Roberts, D. C., Adams, H., Adler, C., Aldunce, P., Ali, E., Begum, R. A., Betts, R., Kerr, R. B., Biesbroek, R., et al.: Climate change 2022: impacts, adaptation and vulnerability, 2022.

Siler, N. and Roe, G.: How will orographic precipitation respond to surface warming? An idealized thermodynamic perspective, Geophysical Research Letters, 41, 2606–2613, https://doi.org/10.1002/2013GL059095, 2014.

Thomas, J.: Impacts of climate change on orographic precipitation - A model study using the Piggybacking method, Master's thesis, Department of Physics and Astronomy, Heidelberg University, 2022.

---

## Author Response (AR2)

**Author's response for egusphere-2022-740, revised submission after editorial decision on 7 January 2023**

We would like to thank the editor and the referees for the assessment of our revised paper. The manuscript submitted for final publication deviates from the accepted version in the following aspects.

Requested changes:

- Modified Fig. 8 and Fig. 9 to make them readable for people with color blindness (added symbols)

- We have uploaded the research data for this manuscript in the repository KITopendata (with doi:10.5445/IR/1000154410) and have modified the "code availability" section accordingly: "ICON model output (2d fields) and post-processing scripts are available for download (Thomas et al., 2023). The full 3d model output fields are available upon request." The 3d model output files are too large (700 GB) to be uploaded onto the repository and will be archived on a tape archive at KIT.

- A reference to the OpenStreetMap licensing conditions has been added in the caption of Fig. 2: "OpenStreetMap data is distributed under the Open Database License (ODbL, see \url{openstreetmap.org/copyright})."

Other minor corrections:

- L271: corrected typo than -> then

- Removed reference to Hall et al. 2012 (as the recorded presentation is only difficult to access because it requires unsupported software)

- Other minor corrections in the reference list (added links etc.)